



# On the energy budget of a low-Arctic snowpack

Georg Lackner[1,2,3,4], Florent Domine[2,3,5], Daniel F. Nadeau[1,4], Annie-Claude Parent[1], Francois Anctil[1,4], Matthieu Lafaysse[6], and Marie Dumont[6]

[1]Department of Civil and Water Engineering, Université Laval, Québec, Canada
[2]Takuvik Joint International Laboratory, Université Laval (Canada) and CNRS-INSU (France), Québec, Canada
[3]Centre d'Études Nordiques, Université Laval, Québec, Canada
[4]CentrEau, Université Laval, Québec, Canada
[5]Departement of Chemistry, Université Laval, Québec, Canada
[6]Univ. Grenoble Alpes, Université de Toulouse, Météo-France, CNRS, CNRM, Centre d'Études de la Neige, 38000 Grenoble, France

**Correspondence:** Georg Lackner (georg.lackner@mailbox.org)

**Abstract.**

Arctic landscapes are covered in snow for at least six months of the year. The energy balance of the snow cover plays a key role in these environments, influencing the surface albedo, the thermal regime of the permafrost, and other factors. Our goal is to quantify all major heat fluxes above, within, and below a low Arctic snowpack at a shrub tundra site on the east coast of
Hudson Bay in eastern Canada. The study is based on observations from a flux tower that uses the eddy covariance approach and from profiles of temperature and thermal conductivity in the snow and soil. Additionally, we compared the observations with simulations produced using the Crocus snow model. We found that radiative losses due to negative longwave radiation are mostly counterbalanced by the sensible heat flux, whereas the latent heat flux is minimal. At the snow surface, the heat flux into the snow is similar in magnitude to the sensible heat flux. Because the snow cover stores very little heat, the majority of
the heat flux into the snow is used to cool the soil. Overall, the model was able to reproduce the observed energy balance, but due to the effects of atmospheric stratification, showed some deficiencies when simulating turbulent heat fluxes at an hourly time scale.

## 1 Introduction

The Arctic winter, characterized by low solar radiation and freezing air temperatures, presents extreme conditions to which
local populations, flora, and fauna must adapt. Climate change imposes additional constraints. Indeed, recent studies have shown that the warming in the Arctic is most pronounced during the cold season (Graversen et al., 2008; Boisvert and Stroeve, 2015), and winter warm spells are becoming more frequent (Graham et al., 2017). This warming, both episodic and perennial, alters the properties of seasonal snow. Since snow is a highly reflective medium with low thermal conductivity, it impacts the entire energy balance at the Earth's surface and leads to repercussions for many fields such as hydrology, permafrost modeling,
weather forecasting, and climate modeling (Meredith et al., 2019).



The energy budget can be calculated at the snow surface using a control surface approach or by considering the snowpack as a whole and therefore relying on a control volume. Using the control surface approach, incoming heat fluxes at the snow surface are counterbalanced by the outgoing fluxes, such that:

$$Q_* + Q_H + Q_E + Q_s + Q_A = 0, \tag{1}$$

where $Q_*$ is the net radiation, $Q_H$ is the sensible heat flux, $Q_E$ is the latent heat flux, $Q_A$ is the flux of advected energy (e.g. by rain falling on the snowpack) and $Q_s$ is the snow heat flux, all in W m$^{-2}$. The fluxes are considered to be positive if directed towards the surface and negative if directed away from the surface. When taking the snowpack as a whole, considering a control volume applied to a horizontally homogeneous snow cover, the energy balance takes the form:

$$\frac{dU}{dt} = Q_* + Q_H + Q_E + Q_A + Q_G, \tag{2}$$

where $dU/dt$ is the rate of change of internal energy of the snow and $Q_G$ is the ground heat flux. Each alternative (equation 1 and 2) has its strengths and weaknesses when it comes to monitoring the energy balance in the field. For example, (1) requires less instrumentation but delivers no information on how the energy is distributed between the snowpack and the ground. While, (2) does provide these details, which are key to permafrost studies. However, this equation requires more instrumentation, which makes it more prone to error.

Measuring all constituents of (1) and (2) in Arctic and Subarctic regions involves many challenges related to field conditions. For instance, the remote locations of the sites greatly complicates maintenance operations, the low solar radiation limits the amount of energy available for stations powered by solar panels, and harsh meteorological conditions (rime, blizzards, etc.) can alter the performance of the instruments. For all these reasons, data for *in situ* snow energy balance in high-latitude environments are very sparse. Moreover, these data tend to suffer from an unclosed energy budget, meaning that the available
energy from net radiation is not equal to the sum of all recorded heat fluxes, and therefore a residual term remains. This phenomenon occurs for almost all energy budget studies and is not restricted to Arctic environments or winter conditions (Foken, 2008).

Of the few existing studies on the winter energy budget in cold regions, a study by Langer et al. (2011) at a polygonal tundra site in northern Siberia found that longwave radiation was the dominant term in the surface energy budget. Langer et al. (2011)
found that 60% of the radiative losses were counterbalanced by the ground heat flux, and the remainder was counterbalanced by the sensible heat flux. Contributions of the latent heat flux were small. They identified cloudiness as the main controlling factor of the surface energy budget. Westermann et al. (2009) reported similar results at a high-Arctic permafrost site, and found latent heat fluxes to be insignificant for the average energy balance. They also characterized atmospheric stratification and found it to range from mostly stable to near-neutral. Lund et al. (2017) measured the energy balance over different surfaces,





such as wet and dry tundra and a glacier, and found only small differences in components of the surface energy budget between the surface types. That study also found that radiation was predominately balanced by sensible and ground heat fluxes.

The above-mentioned studies all show the small contribution of latent heat fluxes to the overall energy budget in the presence of snow. However, according to Liston and Sturm (2004), sublimation in the Arctic can make up as much as 50% of the total winter precipitation. A great deal of uncertainty surrounds this percentage, given the difficulty of measuring both sublimation and solid precipitation, not to mention the large spatiotemporal variation of sublimation in response to varying weather and geographic conditions, such as proximity to water bodies. During blowing snow events, Pomeroy and Essery (1999) reported sublimation rates as high as 0.075 mm water equivalent per hour in the Canadian Prairies. The high rates might explain the high fractions of sublimation of the total winter precipitation reported by Liston and Sturm (2004).

Data on the thermal regime of the snowpack are even more scarce. One of the only studies covering this topic was conducted in the Canadian Prairies by Helgason and Pomeroy (2012). This study found that the measured rate of change of internal energy of the snowpack was systematically lower than the residual of the other energy budget terms (right-hand side of (2)). Thus, Helgason and Pomeroy (2012) were not able to close the energy budget and attributed the remaining energy to "an unmeasured exchange of sensible heat [...] from the atmosphere to the snowpack."

Sophisticated snow models such as Crocus (Vionnet et al., 2012) are occasionally used for climate studies in the Arctic (e.g. Gascon et al. (2014), Sauter and Obleitner (2015) and Royer et al. (2021)) and the Antarctic (Libois et al., 2015). The accuracy of these models at high latitudes has not been evaluated, though large scale bulk properties such as snow depth seem to be simulated fairly well (Brun et al., 2013). However, studies focusing on the ability of these models to simulate the internal physical properties of snow from Barrere et al. (2017) (using Crocus) and Gouttevin et al. (2018) (using the model SNOWPACK (Bartelt and Lehning, 2002)) at Arctic polygon tundra sites found that the observed vertical profiles of snow density and thermal conductivity were not well reproduced by the models. Studies that focus on the performance of these sophisticated snow models when simulating the surface energy budget at point scale are rare. The only studies we found were conducted in alpine regions (Martin and Lejeune, 1998). More generally, very few studies evaluating the turbulent fluxes simulated by snow models are available, despite this process being identified as one of the major sources of uncertainty in snow cover modeling (Menard et al., 2021; Lafaysse et al., 2017).

Here, we attempt to measure all components of the snowpack energy budget at a low Arctic site and compare the observations with simulations from the snow model Crocus. We (i) explore the radiation budget, (ii) compare three years of turbulent heat flux data to examine inter-annual trends and dependencies on meteorological conditions, (iii) compare those observations with model outputs from Crocus, (iv) establish the full energy budget at the snow surface using observations as well as model outputs to assess the relative importance of its components and (v) assemble the energy budget of the entire snow cover.



## 2  Methods

### 2.1  Study site

Our study site was located in the Tasiapik valley near the village of Umiujaq, Quebec, Canada (56°33'31"N, 76°28'56"W),
on the eastern shore of Hudson Bay. The climate is subarctic with a mean annual temperature of $-4.0$°C and a mean annual
precipitation of between 800 and 1000 mm (Lackner et al., 2021). The vegetation at the site consists of a mixture of lichen
(*Cladonia sp.* mostly *C. stellaris* and *C. rangiferina*) and shrub tundra with dwarf birch (*Betula glandulosa*) and other shrub
species (*Vaccinium sp.*, *Alnus viridis subsp. crispa* and *Salix planifolia*) that range from 0.2 to 1 m tall in the upper part of the
valley, and turns into a forest-tundra populated by black spruce (*Picea mariana*) towards the lower part of the valley. In the
study area, the vegetation consists of 20-30% pure lichen-covered surface, the rest being small shrubs (mostly dwarf birch)
with a lichen or moss understory. Permafrost is discontinuous to sparse and is rapidly degrading (Fortier et al., 2011). In the
area surrounding the study site, the soil consists mainly of sand topped by a thin organic litter. The soil organic content varies
between 1.4 kg m$^{-3}$ and 4.3 kg m$^{-3}$ (Gagnon et al., 2019). A more detailed description of the site can be found in Lackner
et al. (2021).

### 2.2  Instrumental setup

A photo of the experimental setup is shown in Figure 1. The setup included a 10-m flux tower equipped with a sonic anemometer
and a $CO_2$/$H_2O$ gas analyzer located 4.2 m above ground (IRGASON, Campbell Scientific, USA) on a 5°- slope with a SE
aspect. The tower also featured sensors for temperature, humidity (model HMP45, Vaisala, Finland), and wind speed and
direction (model 05103, R.M. Young, USA). Total precipitation was measured with a T200B gauge (GEONOR, USA) equipped
with a single Alter shield. Approximately 10 m west of the flux tower, a second station was equipped with a four-component
radiometer (CNR4, Kipp and Zonen, The Netherlands) and an SR50 snow depth sensor (Campbell Scientific, USA).

The snow heat flux was computed using observations collected on a 1-m pole equipped with 18 T-type thermocouples
deployed from ground level up to a height of 0.82 m (see inset in Figure 1). The thermocouples were unequally spaced, with
separations ranging from 2.5 cm to 10 cm. To avoid disturbing the snowpack in cases where the spacing was small, some
thermocouples were deployed with a 90°- deviation from the main axis. Four TP08 heated needle probes (Hukseflux, The
Netherlands) were deployed at heights of 7, 27, 47, and 67 cm above ground. These probes recorded temperature and effective
thermal conductivity according to the measurement principle and data treatment detailed in Morin et al. (2010) and Domine
et al. (2015, 2016). In short, the needle is heated at a constant power (0.4 W m$^{-1}$ in our case), while a thermistor in the shaft
serves as a reference temperature. The temperature difference between these two parts can then be plotted against a logarithm
of the time. The effective snow thermal conductivity $k_{eff}$ is inversely proportional to the slope that is obtained. In our study,
the needles were heated every two days in a 100 s heating cycle, provided that the snow was below $-2.5$°C to avoid melting
and irreversible alterations of the snow structure. Snow temperatures were measured every 5 min (see Supplementary Figure
1). The pole also had 2 thermocouples below the surface of the ground, at depths of 4 and 14 cm, to compute the ground heat



**Figure 1.** Upper panel: Study site with a) the main 10-m flux tower with the eddy covariance setup, b) a precipitation gauge, c) a 2.3-m high mast hosting the 4-component radiometer and d) a vertical pole holding an array of thermocouples and heated needles. Lower panel: Schematic of the study site illustrating the main instruments monitoring energy balance terms. The whole experimental setup is contained within 20 m.





flux. The pole was installed on a patch of *Cladonia* (5 to 10 cm thick yellowish lichen) and data were collected during the winters of 2018-19 and 2019-20.

In addition to the automatic measurements, snow field surveys were conducted in April 2018 and March 2019. In both cases, three snow pits were dug at different locations in the vicinity of the experimental setup to capture the spatial variability of the snow cover. Snow pit data from previous campaigns (Domine et al., 2015) were also included in the analysis. In each snow pit, the stratigraphy was analyzed and profiles of density, temperature, and thermal conductivity were collected. Thermal conductivity was measured with a portable instrument featuring a TP02 heated needle while density was measured with a 100 cm$^3$ box cutter (Conger and McClung, 2009) and a field scale. No field trips to the site were possible in 2020 due to the

COVID-19 pandemic.

Our study focuses on the winters 2017-18, 2018-19, and 2019-20 for evaluating the performance of the Crocus model using observations, and on the last two winters for examining the observed surface energy balance.

### 2.3 Data processing

#### 2.3.1 Turbulent Heat Flux

A detailed explanation of the procedure for obtaining the turbulent heat fluxes from raw eddy covariance data is provided by Lackner et al. (2021). In short, turbulence data were processed using EddyPro® (version 7.0.3; Li-COR Biosciences, USA), a software package that computes fluxes from raw 10 Hz data, while accounting for several corrections and includes a thorough QA/QC procedure. A program called PyFluxPro (Isaac et al., 2017) was also used to remove spikes and erroneous data that persisted despite the EddyPro® processing. Data gaps were filled using ERA5 (Hersbach et al., 2020) reanalysis data and an

artificial neural network procedure (Hsu et al., 2002) with radiation, air temperature and humidity and the soil temperature as the driving data. Mauder et al. (2013) reported errors of 10%-15% for the processed flux data.

#### 2.3.2 Snow heat flux and internal energy

The heat flux $Q_s$ (in W m$^{-2}$) within the snowpack can be calculated using the measured effective snow thermal conductivity $k_{eff}$ (in W m$^{-1}$ K$^{-1}$) from the heated needles and temperatures from the two adjacent thermocouples using Fourier's law:

$$Q_s = -k_{eff} \frac{\partial T}{\partial z}, \tag{3}$$

where the vertical temperature gradient $\partial T / \partial z$ is evaluated using a central finite difference ($\approx \Delta T / \Delta z$). As both $k_{eff}$ and $\Delta T$ are measured automatically, a continuous time series is obtained. We multiplied $k_{eff}$ by 1.2 to correct for the underestimation of $k_{eff}$ by the needle probe method as reported by Riche and Schneebeli (2013). This method was also used to estimate the ground heat flux $Q_G$, but since soil thermal conductivity was not available, it was taken as 1 W m$^{-1}$ K$^{-1}$ (Lu et al., 2018)

for the frozen soil given its sandy texture.





Neglecting any melt or freeze-related processes and assuming a dry snowpack, the rate of change of the snowpack internal energy $U$ is given by

$$\frac{dU}{dt} = \int\limits_0^h \frac{d[T_{snow}(z)\rho_{snow}(z)c_{p,ice}]}{dt}dz, \tag{4}$$

where $h$ is the snow height (in m), $T_{snow}(z)$ is the snow temperature at height $z$ (in K), $\rho_{snow}(z)$ is the snow density at
height $z$ (in kg m$^{-3}$), and $c_{p,ice}$ is the thermal capacity of ice (in J kg$^{-1}$ K$^{-1}$). In (4), internal energy changes associated with fluctuations in the air temperature contained in the snowpack pore space are neglected. As frequent sampling of the snow cover was not feasible, we used the density profile obtained from a snow pit dug once a year and assumed that it remained constant (see Supplementary Figure 2), knowing that this is a very rough approximation. To calculate the rate of change of the internal energy, we discretized the snowpack into 5-cm thick layers, corresponding to the mean separation of the thermocouples. Each
layer was approximated as isothermal, with a temperature equal to that measured at its center. Under these assumptions, (4) then simplifies to

$$\frac{dU}{dt} = \sum_{i=1}^N \frac{\Delta T_{snow,i}\rho_{snow,i}c_{p,ice}}{\Delta t}\Delta z, \tag{5}$$

where $N$ denotes the number snow layers, $\Delta T_{snow,i}$ is the temperature difference between two subsequent measurements of the i$^{th}$ layer, $\rho_{snow,i}$ is the density of the i$^{th}$ layer, and $\Delta t$ is the time step, which in our case was 5 min.
The error of the thermal conductivity can be as high as 29% according to Domine et al. (2016) while the temperature measurements have an error of 0.75%, the standard accuracy of a type-T thermocouple.

## 2.4 ISBA-Crocus Simulations

### 2.4.1 Model Description

Crocus (Vionnet et al., 2012) and ISBA (Noilhan and Planton, 1989; Noilhan and Mahfouf, 1996) are part of SURFEX version
8.1 (SURFace EXternalisée, Masson et al. (2013)), a modeling platform used by Météo-France (http://www.umr-cnrm.fr/surfex/) to simulate water and energy exchanges between the Earth's surface and the atmosphere in both coupled (Numerical Weather Prediction, climate modeling) and offline (avalanche hazard forecasting, hydrology, surface reanalyses) applications. Crocus simulates the snowpack using up to 50 snow layers each defined by thickness, temperature, density, liquid water content, age, and two micro-structural properties (optical diameter and sphericity, Carmagnola et al. (2014)). These properties evolve
according to the physical processes that take place in the snowpack, such as heat exchanges, snow metamorphisms, and snow compaction. Crocus also simulates the radiation budget at the snow surface and the heat exchanges with the atmosphere above the snowpack. The heat flux into the snow cover is the remainder of the turbulent heat fluxes subtracted from the net radiation.





### 2.4.2 Forcing Variables and Model Setup

The ISBA-Crocus model was run in offline mode, meaning it was not coupled to an atmospheric model. Instead, the model was
driven by local observations of the following meteorological variables: air temperature, specific humidity, wind speed, incoming
shortwave and longwave radiation, atmospheric pressure, and (solid and liquid) precipitation rates. These observations have
been collected at the study site since 2012, except for atmospheric pressure, for which data are available from June 2017
onwards. Data from ERA5 were used to fill this gap by correcting for any bias based on the period when both ERA5 and the
observations were available. Note that the sensitivity of the model to atmospheric pressure is known to be very low. Shortwave
and longwave downwelling radiation between mid-December 2019 and mid-February 2020 had to be replaced by ERA5 data
due to a malfunction of the radiometer. Errors of the forcing data varies between 1-3% for temperature and humidity data and
up to 10% for radiation data. The error of the precipitation data is hard to quantify and likely depends on the wind speed.
Kochendorfer et al. (2018) reported a root mean squared error of 0.15 mm for the precipitation gauge used here.

A soil column of 12 m was defined and divided into 20 layers of increasing depth. Following the soil water content analysis
from Lackner et al. (2021), we also adjusted two soil hydraulic parameters, the saturated soil water content and the field
capacity to better match soil moisture observations. The soil composition was set to 95% sand and 5% silt (Gagnon et al.,
2019) and the vegetation was set to 100% shrubs with heights of 40 cm, as this is the dominant vegetation type in the area. To
ensure the equilibrium of soil moisture and temperatures, we initialized the model with a spin-up of 5 years (2012-2017). Since
observations of precipitation were not available before 2015, ERA5 data had to be used for the 2012-2015 period. The default
version of ISBA-Crocus was used for this study, so no interaction between the low vegetation and snow was implemented as
suggested by some studies (Barrere et al., 2017; Gouttevin et al., 2018). We also used the option in Crocus that allows for the
surface to be 100% covered with snow. This was done to ensure that only contributions from snow-atmosphere interactions
were considered. Lastly, we chose not to apply the option that emulates drifting snow.

### 2.4.3 Calculation of the Turbulent Heat Fluxes

In Crocus, the sensible heat flux $Q_H$ is calculated using the aerodynamic approach. As such, it depends on the temperature
difference between the surface $T_s$ and the air $T_a$, the wind speed $U$, as well as the turbulent exchange coefficient $C_H$, as shown
below:

$$Q_H = \rho_a c_p C_H U \frac{T_s - T_a}{\Pi_s - \Pi_a} \qquad (6)$$

where $\rho_a$ is the air density (in kg m$^{-3}$), $c_p$ is the specific heat of air (in J kg$^{-1}$ K$^{-1}$) and $\Pi_s$ and $\Pi_a$ are the Exner functions
for the surface and the atmosphere, respectively.





The latent heat flux follows a similar approach:

$$Q_E = (\chi L_f + L_v)\rho_a C_H U[q_{sat}(T_s) - q_a] \tag{7}$$

where $L_f$ and $L_v$ are the latent heat of fusion and vaporization (in J kg$^{-1}$), respectively, $q_{sat}(T_s)$ is the saturation specific humidity above a flat ice surface at temperature $T_s$ (in kg kg$^{-1}$), $q_a$ is the atmospheric specific humidity (in kg kg$^{-1}$) and $\chi$ is the ratio between the solid and liquid phases of the turbulent mass exchange between the snow surface and the atmosphere.

The turbulent exchange coefficient $C_H$ is discussed in detail in Vionnet et al. (2012) and Noilhan and Mahfouf (1996). In short, this value depends on atmospheric stability, represented by the bulk Richardson number, following the parameterization of Louis (1979).

Lafaysse et al. (2017) details several available parametrizations in Crocus. These were introduced to handle turbulent fluxes under stable atmospheric conditions. According to Lafaysse et al. (2017), the parametrization of Louis (1979) tends to minimize the fluxes under very stable conditions. To address this issue, a threshold on the Richardson number was implemented to maintain a certain level of turbulence in such circumstances. The option of a threshold was initially proposed by Martin and Lejeune (1998) and was used for the simulations performed here, a choice identical to the currently operational model configuration in French mountains.

## 3 Results

### 3.1 Meteorological Conditions

Air temperature, wind speed, and snow height during the three winters are shown in Figure 2. In the fall, the daily mean air temperature is around $-10^\circ$C and drops progressively until January, where it typically varies between $-20^\circ$C and $-30^\circ$C. Starting at the end of February, the air temperatures rise and the first days of positive daily temperatures usually occur around mid-May. For the three winters studied, hourly air temperature excursions above $0^\circ$C occured as early as March and consistently from mid-April. Periods of warmer temperature ($>-10^\circ$C) were occasionally observed in mid-winter. The 2019-20 winter period was marked by more frequent warm spells, making it the warmest of the three winters with a mean temperature of $-14.3^\circ$C (2017-18: $-15.7^\circ$C; 2018-19: $-14.9^\circ$C). Hourly temperatures rarely dropped below $-35^\circ$C and the minimal temperature recorded was $-37.1^\circ$C.

Wind speed shows a similar pattern from one winter to another. In the fall, wind speeds are generally higher due to the temperature gradient between the unfrozen and thus warmer Hudson Bay to the west and the colder surrounding land surfaces. Once Hudson Bay freezes around mid-December, wind speeds drop and display a local minimum in early January. Wind speeds then increase later in the season, to values that are usually lower than in the fall. Only spring 2020 was very windy, with three high-wind events in March and April.





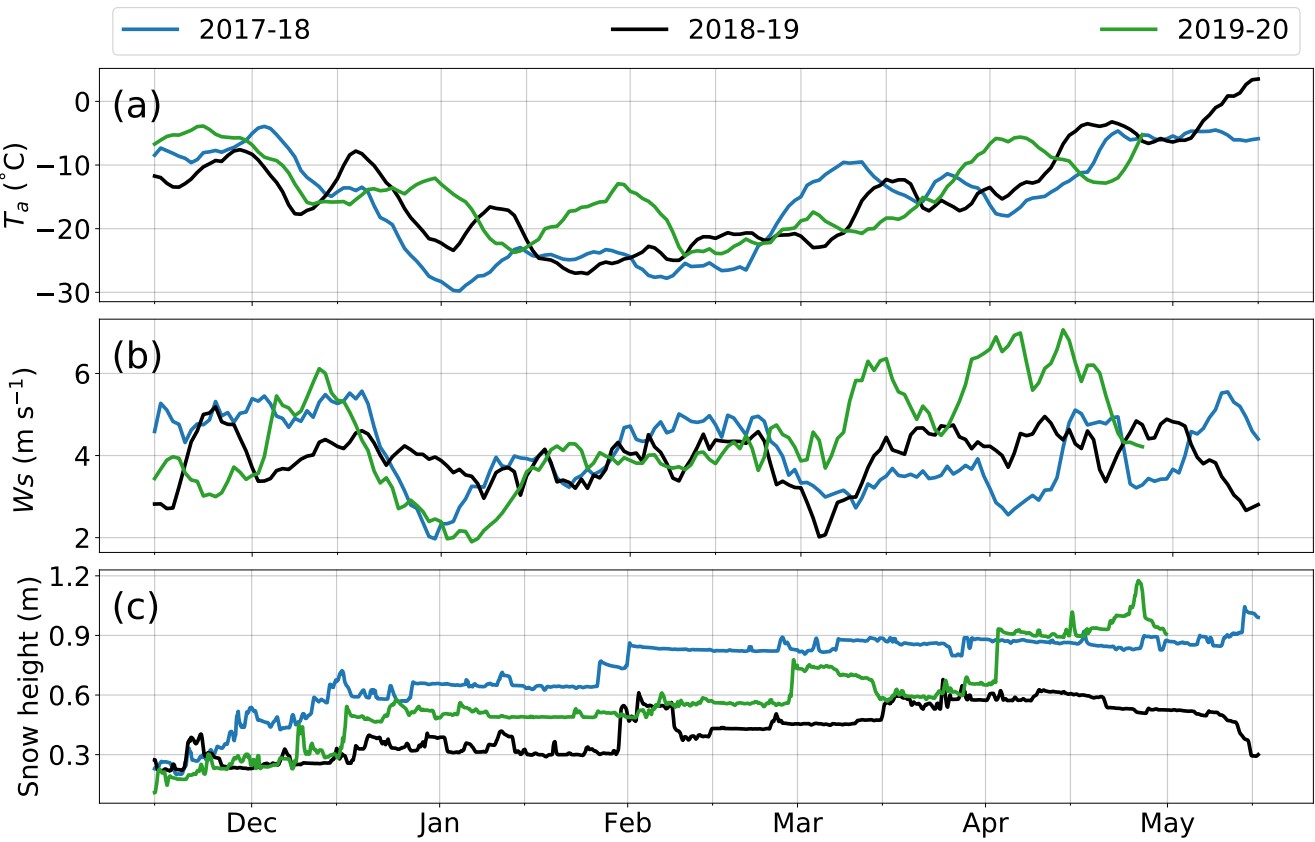

**Figure 2.** Rolling mean of daily air temperature ($T_a$), wind speed ($Ws$) and snow height for three consecutive winters.

Snow usually accumulates quickly in the fall as precipitation events are more frequent due to the large low pressure systems. Just as for the wind speed, precipitation rates drops at the end of December and remain rather low until March, resulting to no substantial changes in the snow height. In the spring, precipitation increases again slightly, leading to a steady increase in snow height so that the maximum values ($\lesssim 1.2$ m) are mostly observed in April.

## 3.2 Surface Energy Budget at the Snow-Atmosphere Interface

### 3.2.1 Observations

**Radiation Budget**

Radiation is the most important component of the energy budget and is highly influenced by the presence of snow. The short and the longwave components are shown in Figure 3, together with the net radiation.

The shortwave radiation depicted in the upper panel in Figure 3 shows one of the most important properties of snow: its high

albedo. This results in net shortwave radiation values close to zero throughout mid-winter, which then started to increase at the



**Figure 3.** Measured radiation budget during the winters of 2018-19 (left column) and 2019-20 (right column). The upper panels show the upwelling and downwelling shortwave radiation, the middle panels show the corresponding long wave components and the lower panels show the short, long and total net radiation.





end of January. However, the net shortwave radiation remained low at the end of winter with maximum daily means of around 50 W m$^{-2}$. The associated albedo varied between 0.82 and 0.92.

The longwave radiation exhibits variations similar to the air temperature variations over the winter, with minimums for both upwelling and downwelling radiation in January and February. However, the difference between upwelling and downwelling
fluxes remained similar, meaning that the resulting net longwave radiation is more constant than its shortwave counterpart. Furthermore, the net longwave radiation was negative, as the upwelling flux exceeded the downwelling flux.

Overall, the longwave radiation dominated the radiation budget in mid-winter and was gradually counterbalanced by an increase in shortwave radiation towards spring. As a result, the total net radiation was mostly negative in the winter and thus removed energy from the snowpack. From early April, the net radiation became positive and thus provided energy to the
snowpack.

**Turbulent Heat Fluxes**

Figure 4 shows the sensible $Q_H$ and latent $Q_E$ heat fluxes for the three winters examined in our study. Both exhibited some short-lived fluctuations, depending on the prevailing meteorological conditions and similar seasonal patterns. $Q_H$ was positive throughout winter except for some brief periods where it was negative at the end of the season. Typically, $Q_H$ varied between
5 W m$^{-2}$ and 15 W m$^{-2}$ from mid-November to mid-March and then slowly decayed until the end of winter. In addition to the general trend of decreasing sensible heat fluxes towards the end of winter, larger variations are particularly visible during this period. These fluctuations can be mainly explained by variations in incoming shortwave radiation and wind speed.

The evolution of $Q_E$ differs from that of $Q_H$ as it shows an increase in fall, remain close to zero in mid-winter with regular occurrences of condensation events (for hourly values), and starts to decrease in March. Thus, from January to early April, $Q_E$
has a negligible contribution to the surface energy budget. Given that $Q_E$ can be translated to sublimation ($Q_E = L_s E$, where $L_s$ is the latent heat of sublimation and $E$ is the water vapor mass flux), very little snow is lost due to sublimation during this period. In the fall and spring, the sublimation rates were higher ($> 0.15$ mm day$^{-1}$) but still rather low, resulting in an average of only 5% of the precipitation being sublimated in the three winters.

Figure 5 presents the sublimation and condensation rates for all three winters examined in this study. Positive values corre-
spond to sublimation while negative values indicate condensation. The rates are mostly very low and close to zero. Only about 14% of the days had sublimation rates higher than 0.2 mm day$^{-1}$. Condensation occurred regularly but values were generally below 0.2 mm day$^{-1}$, with only a few exceptions.

As indicated by equations 6 and 7, when typical modeling approaches are used, both $Q_H$ and $Q_E$ values are expected to be affected by wind speed, but while $Q_H$ is additionally dependent on the temperature difference between the air and the surface,
$Q_E$ depends on the vapor pressure deficit. In order to gain insight into the processes that control turbulent heat fluxes, the dependence of $Q_H$ and $Q_E$ on these variables is depicted in Figure 6.

Both $Q_H$ and $Q_E$ show a rather linear dependence on wind speed. However, while the scatter around the regression line is more limited for $Q_H$, for $Q_E$, the scatter increases with wind speed. Above $\approx 6$ m s$^{-1}$, the dispersion is clearly more pronounced, which is the signature of blowing snow events, as confirmed by visual inspection of time-lapse images. As expected,




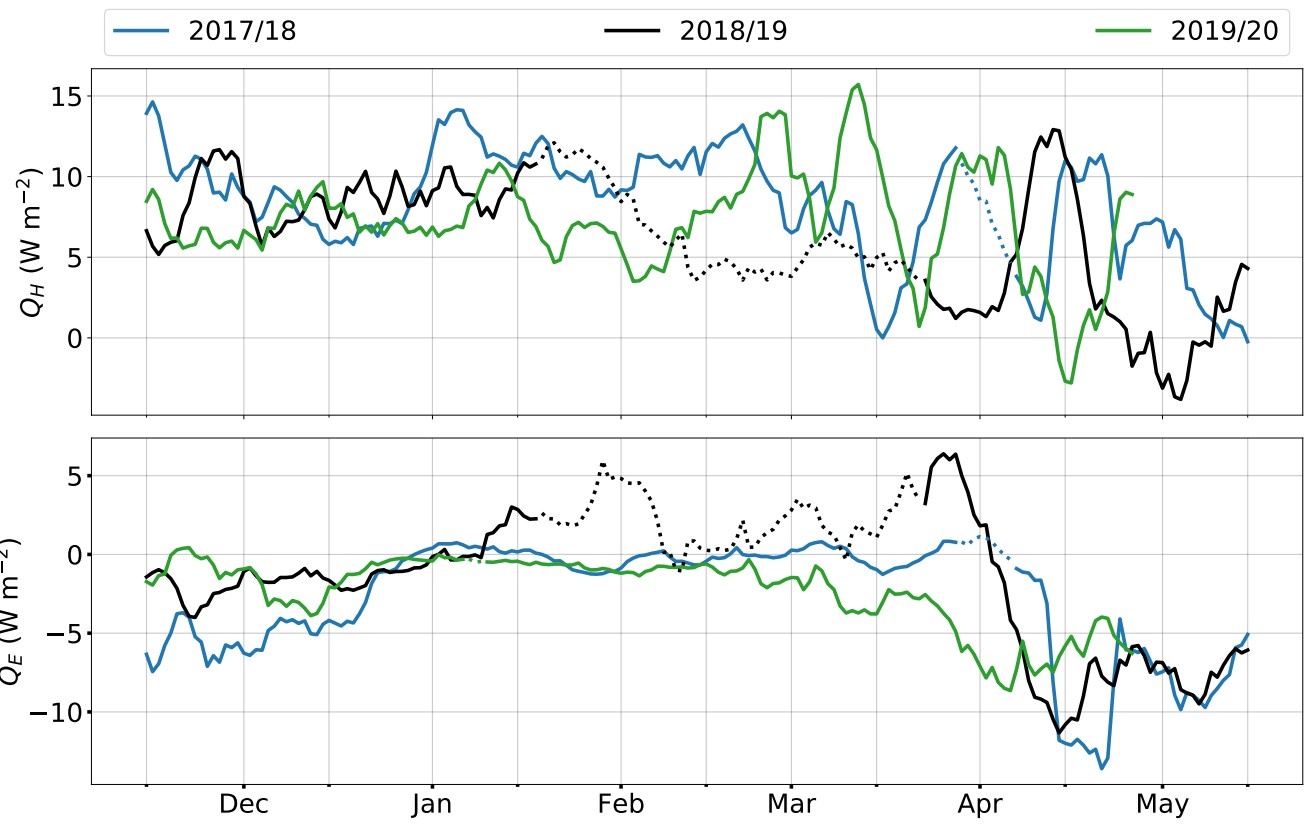

**Figure 4.** Rolling mean of measured daily sensible heat flux ($Q_H$) and latent heat flux ($Q_E$) for the study period. Dotted lines indicate periods of instrument failure which have been gap-filled.

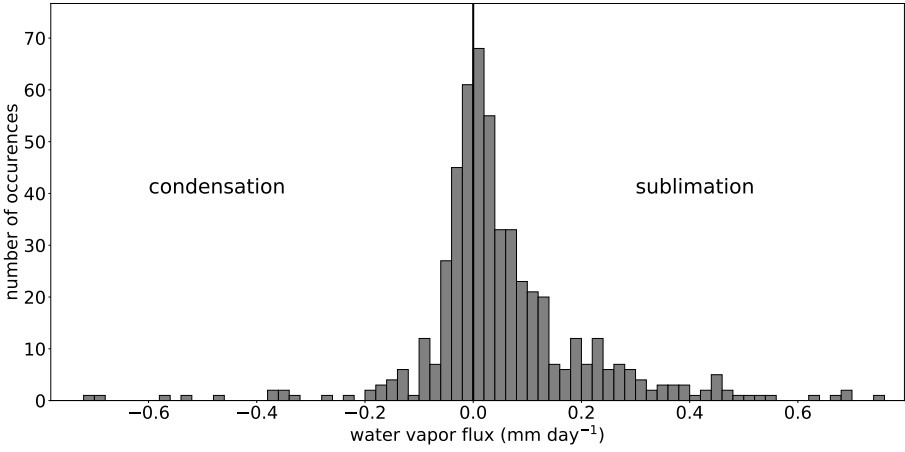

**Figure 5.** Histogram of daily water vapor flux (values $>0$ indicate sublimation and $<0$ indicate condensation) for all three winters.

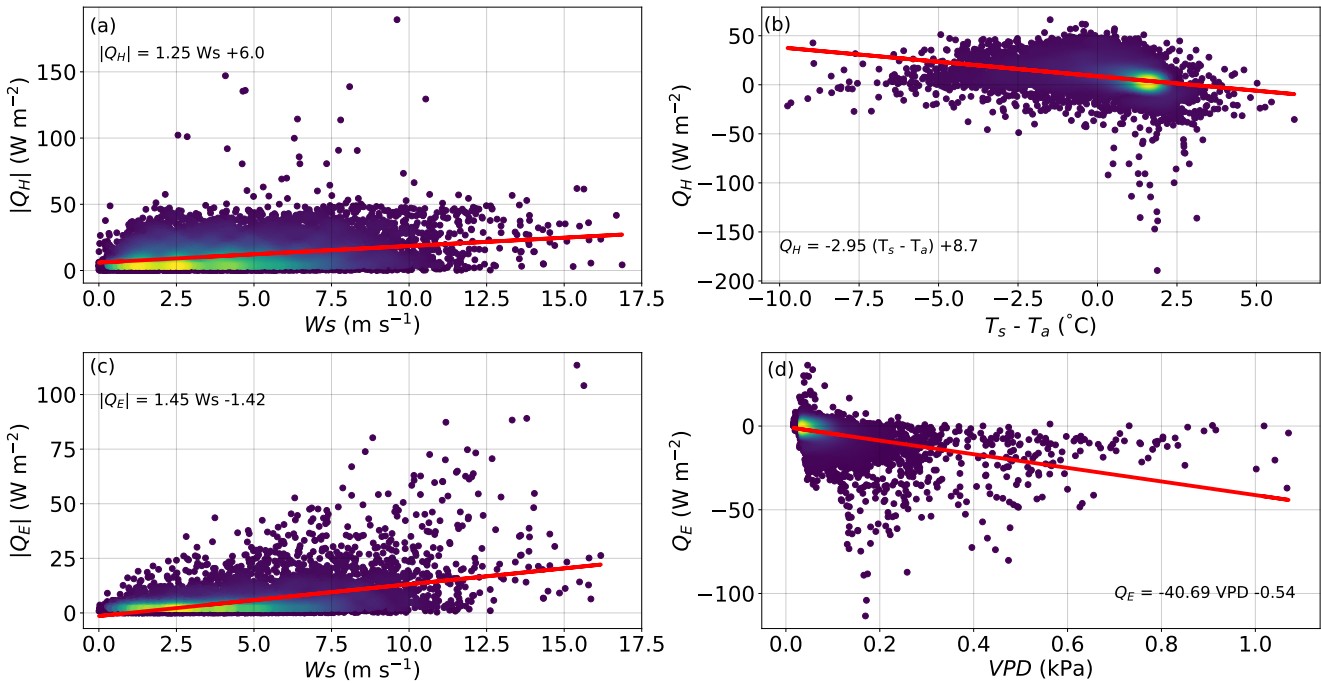

**Figure 6.** Scatter plot of hourly values of sensible ($Q_H$, upper row) and latent ($Q_E$, lower row) heat fluxes against some key meteorological variables: (a) and (c) wind speed; (b) temperature difference between the air and the snow surface; (d) vapor pressure deficit. The color code indicates the density of points, where yellow represents high density and blue low density.

$Q_H$ decreases with higher temperature differences between the surface and the air. However, the lowest values are not observed for the highest temperature differences.

Another important controlling variable for $Q_E$ is the vapor pressure deficit, which states the difference between the amount of water vapor the air currently holds and the amount it can contain when it is saturated. The relationship between $Q_E$ and VPD is more complex than for the previously examined quantities. High vapor pressure deficits (VPDs) do not necessarily result in

high $Q_E$, but the VPD clearly acts as a lower boundary for $Q_E$, as condensation does not occur for higher VPD values. As expected, we notice that condensation episodes ($Q_E > 0$) occur when the air is close to saturation (VPD$\lesssim$ 0.2 kPa).

### 3.2.2 Modeling

**Turbulent Heat Fluxes and Radiation**

Daily means of the $Q_H$, $Q_E$, and $Q_*$ time series simulated by ISBA-Crocus are compared to observations in Figure 7. Both $Q_H$

and $Q_E$ show considerable scatter that is reflected in a relatively small Pearson correlation of 0.67 for $Q_H$ and 0.72 for $Q_E$. The mean bias is rather small with 2.9 W m$^{-2}$ for $Q_H$ and 4.2 W m$^{-2}$ for $Q_E$. The qq-plot for $Q_H$ reveals some discrepancies for high and low values, which are over and underestimated, respectively. For observations in the interdecile range, only a slight overestimation is detectable. The model underestimates $Q_E$ on all the observed values, and more markedly at both ends





of the distribution. $Q_*$ is accurately simulated along most of the range of observed values. Only minor deviations are observed

for very low and very high observed values. The correlation between simulated and observed $Q_*$ of 0.73 is slightly better than

$Q_H$ and $Q_E$, just as the mean bias which is 1.6 W m$^{-2}$.

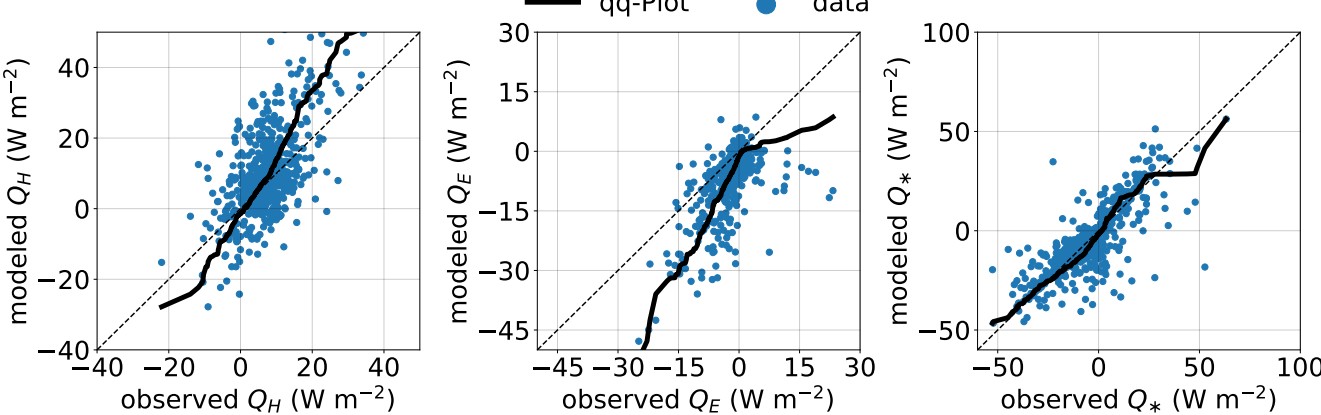

**Figure 7.** Comparison between daily means of modeled and observed $Q_H$, $Q_E$ and $Q_*$ for all three study winters. Overlain is a quantile-quantile plot.

The Pearson correlation between the modeled and observed turbulent heat fluxes is sensitive to the averaging time used. While it drops for $Q_H$ and $Q_E$ when the averaging time is decreased from daily to hourly means (0.53 for $Q_H$ and 0.64 for $Q_E$), it stays constant at 0.73 for $Q_*$ (see Supplementary Figure 3).

**Surface Energy Budget**

A comparison between model outputs and observations for all constituents of the surface energy budget using a control surface framework (equation 1) for a 9-day-period is presented in Figure 8. This period was chosen carefully, to ensure quality data. All instruments functioned properly and the snow cover height was relatively stable at around 55 cm, with one thermal conductivity sensor and two temperature sensors very close to the snow surface (at heights of 47, 49.5, and 52 cm, respectively). The residual

snow heat flux in Figure 8 is obtained by subtracting the heat fluxes from the radiation.

At the surface, net radiation is mainly counterbalanced by $Q_H$, as $Q_E$ is negligible in mid-winter. The diurnal cycle of $Q_S$ very closely follows $Q_*$ (except March 17th). The transition from positive to negative values corresponds to increases in snow temperature, which were highly variable in the period shown (see Supplementary Figure 4). The snow heat flux mostly follows the residual within about 15 W m$^{-2}$ except at the end of the period shown in Figure 8. During that period, the differences

become larger, up to about 40 W m$^{-2}$. At certain moments, a few hours of phase-shifting between the residual and the snow heat flux can be observed. The deviation between the measured snow heat flux and the residual snow heat flux could not be associated with any change in the meteorological conditions and its source thus remains unclear. The mean difference between the residual and measured snow heat flux is 7.7 W m$^{-2}$.





**Figure 8.** Comparison between observed and modeled hourly means of all constituents of the energy budget at the snow surface from March 14 to 23 2020. The residual snow heat flux is obtained by subtracting the turbulent heat fluxes from the net radiation.





Although Figure 8 represents a short time period, it allows for the visualization of the behavior of the model on an hourly
scale. There is a good agreement between modeled and observed $Q_*$, except for the last two days where a deviation of 10 to 50
W m$^{-2}$ separates both curves. $Q_H$ is well simulated at times, but there are periods during which the discrepancy between the
simulated and direct observations is larger (up to 30 W m$^{-2}$) and even includes an error in the sign of the flux (March 18th).
$Q_E$ is also very well simulated but with very little variations during this period, while the mean observed deviation for $Q_S$ is
3 W m$^{-2}$.

**Energy Budget of the Snowpack**

The energy budget at the snow surface can only be analyzed when thermal conductivity and snow temperature measurements
are available very close ($\lesssim 10$ cm) to the surface. As these periods are rather limited, we present the analysis of the energy
budget over the whole snowpack in Figure 9 for both the observations and simulations, over a period of 100 days. The observed
residual in Figure 9 is calculated using only radiation, turbulent heat fluxes, the ground heat flux and the internal energy of the
snowpack according to the equation $dU/dt = Q_* + Q_H + Q_E + \cancel{Q_A}^{\,0} + Q_G + Res.$

On the observation side (Figure 9a), as noted in Figure 3, net radiation is negative during the first few months of the winter.
We note that these radiative losses are largely compensated for by sensible energy input from the atmosphere and heat flux from
the ground. The latent heat flux plays a very modest role in snowpack energy exchange, alternating between periods of heat
input through condensation and periods of heat loss through sublimation. Periods of low net radiation are associated with times
when the residual is significant. Finally, the rate of change of the snow internal energy is low, indicating periods of warming
and cooling of the snowpack, with no clear trend over the period examined. The ground heat flux $Q_G$ represents a substantial
part of the energy budget of the snowpack. In early winter, it is the second largest energy flux after $Q_H$, but towards spring,
$Q_G$ gradually increases and becomes the most important flux, counterbalancing $Q_*$.

The rate of change of internal energy remains very small all throughout the study period and is not equal to the energy re-
maining after subtracting the turbulent and ground heat fluxes from the net radiation. Thus, the unexplained residual constitutes
a significant proportion of the energy balance, with a mean of 6.8 W m$^{-2}$ for the entire period and with maxima of up to 20 W
m$^{-2}$. From mid-November until the end of December, during which the agreement is better, the mean residual is 3.8 W m$^{-2}$.

The simulated energy budget in Figure 9b) compares very well to the observed budget in Figure 9a). The model accurately
simulates the main temporal patterns and in general, follows the observed flux partitioning. As we saw earlier (Figure 7), $Q_*$
is well simulated, but this is slightly less true for $Q_H$ and $Q_E$. The model overestimates $Q_H$ especially during the first and last
months under study. However, $Q_E$ is almost always negative (sublimation) according to the model, while 47% of the days still
show a positive $Q_E$ (condensation). Another striking difference is the role of $Q_G$ in the energy balance. The contribution of
$Q_G$ is much smaller in the model (31% of $Q_*$ on average) than in the observations (47% of $Q_*$).

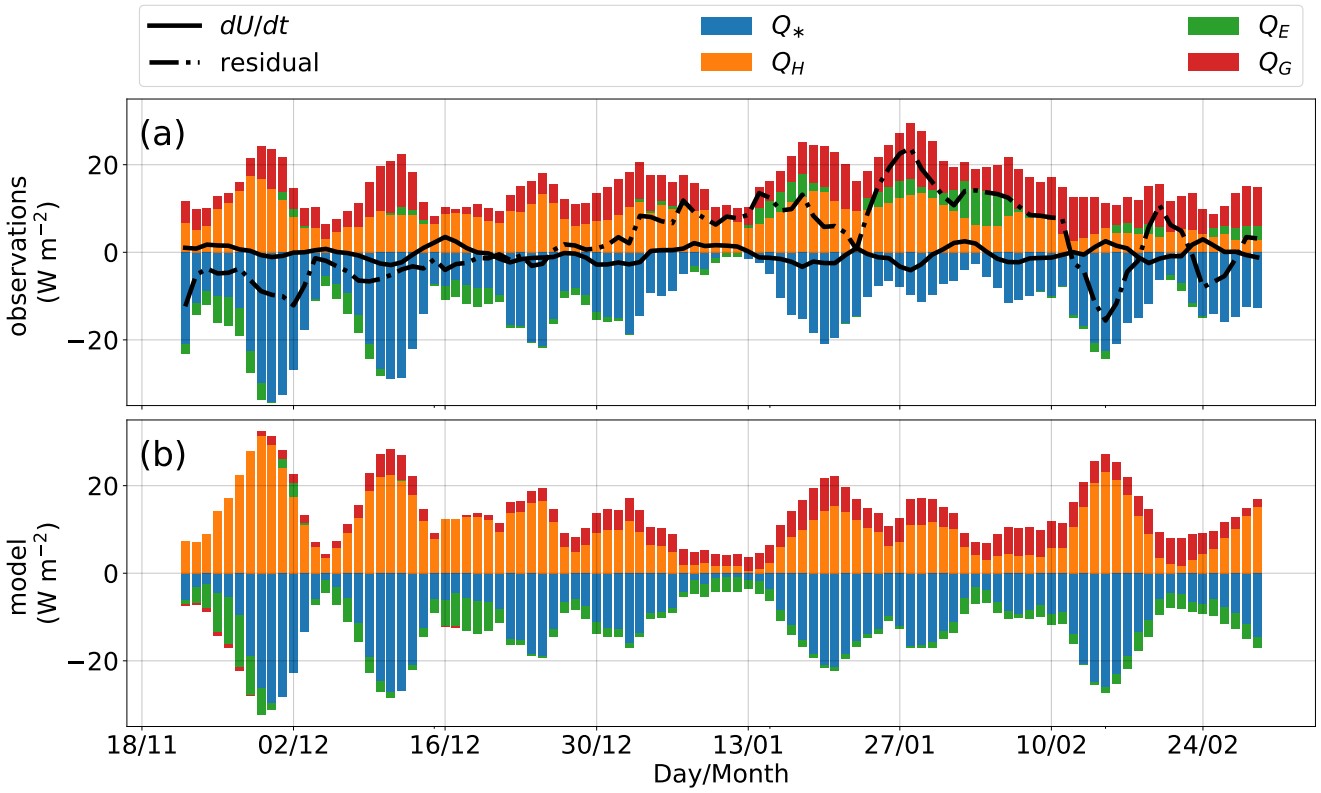

**Figure 9.** a) Observed and b) simulated daily snowpack energy budget terms comprising sensible ($Q_H$) and latent heat fluxes ($Q_E$), net radiation ($Q_*$), ground heat flux ($Q_G$), and the change in the internal energy of the snowpack $dU/dt$, during the first half of winter 2018/19. The modeled ground heat flux also includes the heat storage change $dU/dt$ in the snowpack. The modeled snow enthalpy change is not shown because the modeled enthalpy includes changes due to precipitation and is therefore not comparable to observations.

## 4    Discussion

### 4.1    Turbulent Heat Fluxes and Radiation

The sensible heat flux exhibited a consistent seasonal pattern during all three years (Figure 4) despite varying meteorological conditions. This suggests that there is a consistent limiting factor that prevents large sensible heat fluxes. The energy input would allow for greater $Q_H$ as the site is located south of the polar circle (56°33'N) where there is no polar night. At sites at much higher latitudes in Svalbard (78°55'N) (Langer et al., 2011) and Siberia (72°22'N) (Westermann et al., 2009), the

average sensible heat fluxes were found to have a similar range (between 5 and 15 W m$^{-2}$) to the one identified in this study.

In order to better understand the magnitude of the sensible heat flux, we examine atmospheric stratification, which is usually divided into three turbulence regimes (Steeneveld, 2014; Mahrt, 2014). First, in a very stable boundary layer, turbulence is relatively weak and intermittent, and the flow is dominated by drainage winds and other mesoscale motions. Second, in a weakly stable boundary layer, the atmosphere remains stably stratified (the potential temperature increases with height), but





turbulence becomes the dominant type of transport. Thirdly, in an unstable boundary layer, the potential temperature decreases with height and turbulence is fully developed. We calculated the bulk Richardson number $Ri_b$ between the snow surface and the measurement level of the eddy covariance (Table 1) to identify the dominant atmospheric regimes at our site. The stratification was found to be mostly weakly stable ($0 \leq Ri_b \leq 0.25$) or stable ($Ri_b > 0.25$), with unstable stratification ($Ri_b < 0$) making up only 22.1%. For 66% of the time, the boundary layer is weakly stably stratified, meaning that buoyancy acts to limit the

magnitude of the turbulent fluxes.

| unstable | weakly stable | stable |
|----------|---------------|--------|
| 22.1% | 66.5% | 11.4% |

**Table 1.** Distribution of atmospheric stability based on the bulk Richardson number $Ri_b$, where $Ri_b < 0$ indicates unstable conditions, $0 \leq Ri_b \leq 0.25$ implies weakly stable conditions and $Ri_b > 0.25$ indicates stable conditions.

Atmospheric stratification likely influences the capacity of the model to simulate the turbulent heat fluxes. The model accurately reproduced the observed turbulent heat fluxes on large time scales (daily and longer periods) while showing larger errors for sub-daily periods. The most probable reason for the degradation associated with shorter timescales is the increasing influence of atmospheric stability on the heat fluxes. Modeling turbulent heat fluxes under stable atmospheric conditions is thus

clearly more complex. There are three parametrizations available (Lafaysse et al., 2017) in Crocus that handle atmospheric stability. One of these sets a limit to the bulk Richardson number and thereby maintains a minimum level of turbulent mixing even for stable conditions. In this study, we noticed only small differences between these three options ($<1$ W m$^{-2}$ at the daily scale), whereas major differences were reported from Col de Porte, France, as well as at almost all the sites in the ESM-SnowMIP dataset (Ménard et al., 2019). This is probably because our site is windy, which helps to keep $Ri_b$ low and away from

the very stable conditions that are more difficult to handle. Nevertheless, our results objectively confirm the statement of e.g. Menard et al. (2021) that turbulent fluxes are by far the most uncertain component of the energy balance simulation in snow cover modelling. We therefore highly recomnd future complementary evaluations of turbulent fluxes of snow cover models, as was done in this paper for various climate and environments. Meanwhile, further studies are underway at the site targeted in this study to evaluate optimal turbulent heat flux parametrizations.

In addition to the atmospheric stratification, VPD is another important factor impacting $Q_E$. At our site, we observed very low $Q_E$ values, coinciding with periods of intense cold ($\lesssim -15^\circ$C) when the atmosphere can accommodate a very limited amount of water vapor. We hypothesize that moist air advected from Hudson Bay might further reduce the VPD and thus $Q_E$, as westerly winds were frequently observed and Hudson Bay remains unfrozen until mid to late December. These two factors, combined with the prevalence of stable atmospheric conditions, likely explain the fact that sublimation accounts for only 5%

of winter precipitation, while other studies (Pomeroy and Essery, 1999; Liston and Sturm, 2004) have reported much higher percentages (10-50%) in other locations in the Canadian Arctic. However, sublimation during blowing snow events, which frequently occur at the site (observed several times per week on time-lapse images), could be substantially underestimated for two reasons. First, snow particles can obstruct the optical path of the infrared gas analyzer and cause a malfunction. Thus, $Q_E$ observations during these events are marked by greater uncertainty. Second, snow particles during blowing snow events can





be lifted several meters in the air and thereby above the eddy covariance system, which is installed at 4.2 m above the ground. Snow particles that sublimate above the instrument cannot be measured, which then leads to an underestimation of the total sublimation rate. Yet, Mann et al. (2000) measured snow particle density profiles during blowing snow events in Antarctica and showed that the decrease in snow particle density can be approximated by a power law. This suggests that the fraction of blowing snow sublimation above the flux system is most likely rather small. The authors also showed that the air can become

almost fully saturated within several meters above the surface, which then acts to reduce the sublimation rate during blowing snow events. Altogether, there are still many unknowns with snow sublimation measurements and these unknowns may be the source of many possible errors. Therefore, there is considerable inherent uncertainty associated with measurements of the total amount of snow that is sublimated in a winter.

    Although it suffers from a lack of evaluation, a parametrization of the sublimation of blowing snow from Gordon et al.

(2006) was implemented in Crocus by Vionnet et al. (2012). However, we did not apply this option and thus the simulations from Crocus only include the sublimation from the surface. We chose to omit the blowing snow simulation because most occurrences of blowing snow coincided with instrument malfunctions and no measurements were recorded.

    Similar to other sites with comparable climate conditions, longwave radiation was the main component of the radiation budget, with shortwave radiation becoming more substantial in spring. The mean albedo of 0.85 was also in the expected range

of what is typically found for snow that is largely free of surface impurities (Warren, 1982; Gardner and Sharp, 2010).

### 4.2  Energy Budget

Over a short period of 9 days (Figure 8), the snow heat flux was found to be of the same order of magnitude as the turbulent fluxes, and thus plays a vital role in the surface energy budget. Meaningful measurements of the snow heat flux are difficult to obtain, as various limitations of the measurements have to be overcome, including the limited spatial representativity of

the snow thermal measurements and the frequent high winds which erode and accumulate snow, resulting and rapid spatial changes in snow height. Furthermore, wind pumping, the forced airflow through the upper layers of the snowpack (Colbeck, 1997), and solar radiation can disturb snow heat flux measurements. Thus, we advocate that the control surface approach should be restricted to short-term studies but under favorable conditions (stable snow height and less stable atmospheric conditions). The measured snow heat flux has the same magnitude as the residual snow heat flux, thereby confirming that no major energy

flux is missing.

    For a long-term energy budget study, the control volume is better suited as it does not require thermal conductivity close to the surface. However, the closure issue is worse. Firstly, this method requires temperatures to be measured close to the surface and thus comes with the same challenges mentioned above. Secondly, the method relies on the snow density profile, which could not be regularly measured in this study (only once per year towards the end of winter), as the site is very remote. Additionally,

the density profile is spatially variable. We however estimate that the error for density is not very large ($\lesssim 50 \, \text{kg m}^{-3}$) based on density measurements from earlier in the winter in previous years. Assuming this error in our calculations, the corresponding error in the heat change is presumably less than $0.5 \, \text{W m}^{-2}$ per 5 cm layer. Thus, inaccurate density estimates very likely contribute to the energy imbalance, but they cannot be the main cause. Modeled density profiles from Crocus were tested but





the error for density, particularly in the upper part of the snowcover, exceeded those taking a constant profile. Helgason and

Pomeroy (2012) attributed part of their residual to the fact that the boundary layer might have decoupled from the surface and thereby "rendering the measured eddy-covariance fluxes unrepresentative of the true surface heat fluxes". Although this might be true for some of the measurements presented in our study, based on the fact that snow heat fluxes and residuals are of the same magnitude when looking only at the surface energy budget, we believe that the residual is mostly due to measurement uncertainties.

The ground heat flux is also likely to contribute to the unbalanced energy budget because unfortunately, our measurement setup did not allow us to calculate the storage of heat in the top layer of the soil ($\approx$4 cm). However, the heat flux at the bottom of the snowpack ($\approx$7 cm) was measured using the same principles as the heat fluxes at the surface and the agreement between this flux and the ground heat flux increases our confidence that the measured ground heat flux does not have substantial errors. As the heat change of the snowpack is very small, the majority of the radiation that is not counterbalanced by the turbulent heat

fluxes above the snow surface is used to cool the soil.

Simulations of $Q_H$ and $Q_E$ greatly affects the simulated heat fluxes into the snow and ground because heat fluxes into the snow are calculated as a residual in the model. This can be seen in Figure 9, at the beginning of the period illustrated, until approximately the end of December, when $Q_H$ is overestimated and consequently, $Q_G$ is underestimated. In the second half of the period, the situation changes, and $Q_H$ in the simulation has a similar magnitude as in the observations and the estimates of

$Q_G$ becomes closer to that of the observations. For this reason, a proper simulation of $Q_H$ and therefore also of the atmospheric stability is crucial for the simulation of ground temperature and permafrost melt.

### 4.2.1 Comparison to Previous Modeling Attempts

Studies comparing the full simulated and observed energy balance of an Arctic snowpack are sparse. Westermann et al. (2016) used CryoGrid3 (https://github.com/CryoGrid) to simulate the energy budget of a polygon tundra site in Siberia. They reported

that the model satisfactorily reproduced the winter energy balance but also found that it underestimated all heat fluxes. Helgason and Pomeroy (2012) used the SNTHERM model in the central Canadian Prairies. They found that the latent heat flux and $dU/dt$ were well simulated while the sensible heat flux was overestimated. However, they concluded that the sensible heat flux was undermeasured and that the modeled flux was more applicable.

## 5 Conclusions

In this study, we investigated the energy budget of a low-Arctic snowpack over three winters. Firstly, we analyzed the evolution and inter-annual changes of its constituents, which mainly consited of radiation and turbulent heat fluxes. We then compared observations with the simulated time series from the ISBA-Crocus land surface model. Secondly, during two selected periods, we compared observed and modeled fluxes using a control surface at the snow-atmosphere interface and a control volume that encompassed the snowpack.





In line with previous studies, we found that the sensible heat flux was far superior to the latent heat flux, and provided heat to the snowpack during most of the winters. The sublimation rates were rather low ($<0.4$ mm day$^{-1}$) and made up only around 5% of the winter precipitation which is low compared to some studies that have reported up to 30% and more. We hypothesize that this is mainly due to cold air and that the associated low vapor pressure deficit and sublimation during blowing snow events are possibly undermeasured.

At the surface, net radiation was counterbalanced about equally by the sensible heat flux and the heat flux in the snow. For the control volume, the ground heat flux was found to be the most important heat flux after the sensible heat flux, meaning that there is little energy stored in the snowpack. The imbalance of the energy budget was more apparent when considering the whole snowpack, so we concluded that a large portion of the imbalance was due to errors associated with the heat fluxes in the snowpack.

Overall ISBA-Crocus was able to simulate the main components of the energy budget reasonably well, especially considering their small magnitudes. The turbulent fluxes had much larger errors than the radiative fluxes. The model showed a particular weakness when simulating fluxes under stable atmospheric conditions, which resulted in a decrease in performance from a daily to hourly scale.





*Author contributions.* GL, DN and FD designed research. GL, DN and ACP deployed and maintained instruments. GL and FD performed the field work. GL analyzed data with inputs from DN and FD. GL set up the snow model Crocus with help from ML and MD. GL wrote the paper with inputs from DN and FD and comments from FA, MD and ML.

*Code availability.* The source files of SURFEX code are provided at the Git repository (http://git.umr-cnrm.fr/git/Surfex_Git2.git, last access: 20 June 2021) with several code management tools (history management, bug fixes, documentation, interface for technical support, etc.). Registration is required; a description of the procedure is described at https://opensource.cnrm-game-meteo.fr/projects/snowtools/wiki/Procedure_for_new_users (last access: 25 July 2021).

EddyPro® is available on the Licor website (https://www.licor.com/env/support/EddyPro/software.html) together with the corresponding manual.

PyFluxPro is provided in the Git repository of OzFLUX (https://github.com/OzFlux/PyFluxPro; last access: 25 July 2021). The different levels are described by Isaac et al. (2017).

*Data availability.* Data available upon request from the authors.

*Competing interests.* Florent Dominé and Marie Dumont are Editors for The Cryosphere.

*Acknowledgements.* This study was funded by Sentinel North, a Canada First Research Excellence Fund, under the theme 1 project titled, "Complex systems: structure, function and interrelationships in the North". We are grateful to Denis Sarrazin for technical support of the CEN weather stations and the community of Umiujaq for granting permission to conduct our research. CNRM/CEN is part of Labex OSUG@2020 (ANR-10-LABX-0056). M.D. has received funding from the European Research Council (ERC) under the European Union's Horizon 2020 research and innovation program (grant agreement No 949516, IVORI).



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
