# Peer review of "On the energy budget of a low-Arctic snowpack"

_The Cryosphere, 2021_

## Referee Comment (RC3)

**Review of:  On the energy budget of a low-Arctic snowpack**

**For "The Cryosphere"**

Authors: Georg Lackner, Florent Domine , Daniel F. Nadeau , Annie-Claude Parent , Francois Anctil , Matthieu Lafaysse , and Marie Dumont

Overall this paper should be published. There are earlier papers looking at snow energy balances in the subarctic/low arctic (I refer to Anthony Price' early work in the Schefferville area when he was a PhD student at McGill). In the Lackner et al paper there are notable discrepancies between the real world and the model. Though I thought the discussion was quite good it might be an idea to add a few words directed at the model and why it doesn't seem to model Qh well. There is acknowledgment of this but I wondered if the authors, after their experience with this data set have suggestions to better the model? I have made a few comments below some out of curiousity and others more specific. In tundra environments herbs and shrubs in the snowpack can play a role in the energy balance at I assume they can…especially in the late winter when the sun is becoming more intense and in the early spring as they absorb radiation (I understand coniferous plants can photosynthesize under the snow surface).

Overall I would accept the paper with some minor modifications.

Page 3: line 50: they measured ground heat flux under the glacier? Is that right?
 Line 53: small point: would be slightly clearer to say that 50% of the winter precip is **lost** to sublimation
 Line 59: though in the subarctic, the earlier study by Price (PhD at McGill) was a detailed energy budget of a snowpack. (Water Res. Res Vol12:4: 686-694
Page 7: line 145 or so: as the density of the snowpack changes the amt of air space changes…would this not have an impact on calculating the heat capacity of the snowpack?
Page 8: Line 179: 12 m or 1.2m? earlier you state that you have thermocouples at 4 and 14 cm…so not sure what you are doing here…assume this is a typo
 Line 180: did you measure field capacity in the lab
 Line 194: a small question (though it probably makes little difference) do you adjust specific heat wrt temp – I assume you are but would it make much difference? Thinking here too about air in the snowpack
Figure 2: Probably quite explainable…however there are places here where your snowpack

drops significantly over what appear short periods of time…wind? (coupled with compression) -for example 2018-19 late Jan there is a snowfall (i assume) and on or about 7 FEb there is a sudden drop of snowpack height from about 55cm to 40cm (or so)/ as well in 2019-20 late Feb snowpack is about 75cm or so and by mid March around 60cm…significant wind at this time..assume it is wind scour

p10. Line 255: I assume that this pattern of precip is tied somewhat into the proximity of Hudson's Bay….does the drop in precip in December tie into ice covering a large part of the bay?

Figure 3: net radiation in 2018-19 in early January show a slightly  positive balance out of curiousity what is happening here; similarly 2019-20 in early March; in both cases longwave in and out is balanced …significant cloud cover? Thinking that in years ahead with more of Hudson bay staying open longer there may be increased cloud cover…might be interesting to speculate how this may play a role in the energy budget of these low Arctic snowpacks?

Figure 6: you are inferring a linear relationship here…is there any point? are  these relationships significant? What might be interesting is to look at (for example) in (b) at Ts-Ta 1°C to about 2.3°C the range of Qh is very large, though focused primarily between ~+50 and -45 W/m2…for more or less similar Ts-Ta values you get a very large range of Qh: is there anything of interest here: similarly for (d) between ~.15VPD and .18 VPD a very large Qe range

P 14 line 280: are these relationships statistically significant? Though when looking at these relationships the important thing is the visual message that the model in Qh and Qe under/overestimates for a reasonable range of the W/m2 range

P15 Line 301-302

You refer here to residual snowpack heat flux. In our experience in subarctic and low arctic snowpacks there can be a notable concentration of coniferous shrubs that absorb energy and appear to be photosynthesizing (we were not measuring this but colleagues mention this goes on). Is the snowpack in any way impacted by energy absorption by shrubs in the snowpack at all? Seems it might…of course depending on the characterisitics and density of the shrubs. I see no mention here of shrubby veg…so assume this isn't the case here?

P 15 line 294: would the sensors be in any way impacting the energy balance? as they are close to the surface of the snowpack/ what about blowing snow along the surface…impact the ability of the sensors in any way?

Figure 7. so with Qh you have wide scatter in the model. might be interesting to look at some observed values of Qh and investigate the very large range of modelled responses…to isolate what other variables might be playing a role in producing a large range/

for example at ~+10W/m2 (observed) you have a range of modelled responses between ~ -5 and +35 ….in the upper range of the modelled responses versus the lower range were there any standout differences in other variables (wind speed?)

Figure 7 and Figure 8
I see why you have included Fig 8 but there are some interesting differences between these figures which leads to a couple of questions. 1. In Fig 8 where you are confident you have good control on all measureable variables there is very good agreement between the modelled and observed Qe. However in Fig 7 there is consistent underestimation (I think I am reading this correctly). So what did you learn (and what could we learn) from this discrepancy with respect to the model and where the issues are?
2.With respect to Fig 7 if it wasn't for a number of values >25 W/m2 or below -5 W/m2 the relationship could well be close to vertical as I would guess 90+% of the values fall within the large lump. In Fig 8 the model again overestimates under very good controlled conditions. Any idea why the model seems to overestimate sensible heat? Any physical explanation?

P17. Line: 315: I assume crossing out the Qa term simply means that while you identify it is important it is not included here…the arrow to the zero?

P17 Line 326: so you have more energy coming in than can be accounted for in heat loss from the system/ possibilities? (a bit of guess work here)underestimating energy used to raise the temperature of the ice in the snowpack to melting point? Early in the melt as melting water infiltrates the frozen ground (is it?) the runoff refreezes complicating the issue? Some issue with calculating heat loss from the snowpack re: turbulent fluxes…..and going out on a limb here but are there periods when you may have laminar flow and underestimating heat loss// later I see that isn't an issue…but thought I would leave this comment in

P17 line330: the phrase: "*slightly* less"… are you being generous using this term especially for Qh? re: fig 7

P17 L333: issue re: observations of instrument location an issue…
P 19: line spelling of *recommend*
P20 Line 397: again, if herbs and shrubs are present are they playing a role here?
P21 line 436: spelling of word: *consisted*

---

## Author Response (AR1)

**Responses to Reviewer #1**

**Overall Comments:**
The authors have collected a nice dataset and have produced a potentially very informative and useful paper, examining the snowpack energy balance in a low-Arctic snowpack. I appreciate the difficulty of working in such an environment and I believe that the measurements and modelling have the ingredients for a paper that warrants publication. However, I think there needs to be some additional analysis in order to obtain the best possible interpretation of the data. […] There are not a lot of comprehensive energy budget studies on low-Arctic snowpacks and so making the most use of the data would result in a more useful paper.

Thank you for your encouraging words. We highly appreciated your comments, which clearly helped us improve our manuscript. Please find our answers to your comments below.

1. Some of the time series presented would be much easier to interpret if temperatures were included. Important temperatures would include, air temperature, snowpack surface temperature, average or bulk snowpack temperature, and soil temperature at 4 and 14 cm depth. Soil temperatures would provide an indication of when the soil water was freezing.

We agree with you and added the requested time series. Please see our response to comments 18 and 19.

2. It would be helpful to be able to either see the separate shortwave and longwave radiation balances, or to augment the net radiation with observed and simulated albedo values.

Please see our answer to comment 18.

3. The authors need to state whether the precipitation data were corrected for gauge undercatch and if so, describe the procedure. The precipitation data cannot be used to force a model in their raw, uncorrected state.

Please see our answer to comment 15.

4. Some time series of simulated SWE (with points for late winter observations) and simulated and observed snow depth would be interesting. Since there is no mid-winter melt happening, I am interested in knowing how the observed snow depth time series are affected by snowfall events, whether some depth increases are caused by snowfall or drifting, and whether depth decreases are caused by drifting, sublimation or settling and wind packing. Definitive answers may not be possible but evidence in the data may produce some answers. There are not a lot of comprehensive energy budget studies on low-Arctic snowpacks and so making the most use of the data would result in a more useful paper.

We understand very well your curiosity to learn more about other aspects of Crocus modeling, which have more to do with the properties of the snowpack itself, and not only the energy balance simulation.

In fact, there is so much to say about this that we have compiled all our analyses in second paper that will be submitted shortly to *The Cryosphere,* and that will echo this one. In connection with comment 12, we agreed that it is relevant to compare an observed density profile with a simulated one, and presented some answers to these questions, but which will be substantiated in the upcoming paper.

**Specific Comments:**

5. Line 7-10: I find the following a little confusing: "At the snow surface, the heat flux into the snow is similar in magnitude to the sensible heat flux. Because the snow cover stores very little heat, the majority of the heat flux into the snow is used to cool the soil." I understand that the sensible heat flux is usually downward and I assume that the heat fluxes calculated from the temperature gradients near the top of the snowpack showed a similar heat flux. However, I find "the majority of the heat flux into the snow is used to cool the soil" to be confusing. A downward sensible heat flux into the snow would not cool the soil. The upward soil heat flux into the snowpack would cool the soil. I would reword this part.

This wording is confusing indeed. We made the following modifications to the sentence (in bold):

"[...] the majority of the **upward** heat flux **in** the snow is used to cool the soil."

6. Line 15: One could surmise that the flora and fauna as well as the local populations have adapted to the conditions, which is why there is such concern about changes to the environment affecting the flora, fauna and the traditional way of life of the local inhabitants.

We modified the sentence to illustrate the dynamic nature of these changes and the constant search for equilibrium that ensues (modifications in bold):

"[...] conditions to which local populations, flora, and fauna **are adapting**."

7. Equations 1 and 2: If Qs is derivable for equation 1, it could inform the results from equation 2.

Yes, deriving $Q_s$ from equation 1 can inform the results from equation 2. However, $Q_s$ is further partitioned into the rate of change of internal energy of the snow $dU/dt$ and the ground heat flux $Q_g$, for which equation 2 is still needed.

8. Line 40-42: I agree that lack of energy balance closure in eddy covariance systems is not restricted to Arctic environments or winter conditions. However, the ability of eddy covariance systems to measure fluxes has been documented in many papers as being severely limited under periods of low wind speed and strong stability, which damps turbulence. The prevalence of such conditions may therefore affect the degree to which observations at a given site are affected, even if the underlying mechanisms are the same. Was energy balance closure worse under clam, stable conditions? Figure 1 appears to show a ridge close to the site and I wonder whether drainage flows are affecting the energy imbalance because of the topography.

As we illustrate in Table 1, the cases where the atmosphere is stable ($Ri_b > 0.25$) are relatively marginal (11.4% of the time), as the site is rather windy. As such, a clear trend of lower energy balance closure during calm wind conditions has not been observed. Even though the eddy covariance system is a

source of uncertainty, the other heat flux measurements (e.g. in the snow and ground) are associated with an uncertainty similar to if not greater than the eddy covariance system itself during the winter conditions presented here. Thus, a lower closure not only arises just from problems with the eddy covariance system and thus, there is no clear relation with calmer wind conditions.

We recently published a paper looking at the summer energy budget at this site (Lackner et al. 2021). We indeed observed a clear drop of the energy budget closure when the winds came the ridge, probably because we then find ourselves in a recirculation zone that is hardly compatible with the assumptions underlying the application of the EC approach.

9. Line 95: The authors should specify that the CO2/H2O analyzer is an open path system which may experience interference from snow and blowing snow.

Good point, we expanded on this (modifications in bold):

"The setup included a 10-m flux tower equipped with a sonic anemometer and a $CO_2$/$H_2O$ gas analyzer located 4.2 m above ground (IRGASON, Campbell Scientific, USA) on a 5°- slope with a SE aspect. **Due to the open-path nature of the EC sensor, it was subject to interference in the presence of precipitation and during blowing snow events.**"

10. Line 111: Is a 10 cm spacing of thermocouples starting at -4 cm sufficient to compute the ground heat flux accurately?

Indeed, more refined spacing, as well as deeper soil measurements would have been desirable. For example, in a study at the same site but covering the snow-free period, we estimated the ground heat flux using three ground temperature measurements, at a station some 15 m from the one used here (see Lackner et al. (2021)). Even though there are only two measurement levels, they have the advantage of being directly below the snow temperature profile measurements, which ensures consistency in our analysis. Also, as mentioned in the manuscript, we measured the heat flux at the very bottom of the snow cover and compared it to the ground heat flux, and we found that both fluxes agreed very well. This suggests that the method we use is sound.

11. Lines 125-130: Was coordinate rotation applied to the eddy covariance data to account for the slope? A brief summary of procedures for processing and QA/QC would be informative and useful. Were any u* thresholds applied?

Indeed, we applied the double rotation method to the raw turbulence data. The planar fit method was tested as well, but yielded occasional large spikes and unrealistic results. We surmise that the presence of snow cover constantly changes the surface, making it impossible to use the planar fit method. The processing routine with EddyPro handles an important part of the QA/QC processing, the rest being done by the PyFluxPro program. As for the u* threshold, no filter was applied to the turbulent fluxes in order to maximize the amount of data available for the analysis. To make this information more obvious, we added the following information (in bold):

"A detailed explanation of the procedure for obtaining the turbulent heat fluxes from raw eddy covariance data is provided by Lackner et al. (2021). In short, turbulence data were processed using EddyPro® (ver- sion 7.0.3; Li-COR Biosciences, USA), a software package that computes fluxes from raw 10 Hz data, while accounting for several corrections, **including the application of a double rotation on the raw data to align the coordinate system with the current snow surface. EddyPro**

**also** includes a thorough QA/QC procedure. A program called PyFluxPro (Isaac et al., 2017) was also used to remove spikes and erroneous data that persisted despite the EddyPro® processing."

12. Line 147: What does the model suggest for the snowpack density evolution? How much did it vary from year to year in the snow pits and in the models? I see later that the error is considered greater than assuming a constant density but the sign of the error and reasons are not discussed.

Snow models such as Crocus intrinsically have difficulty simulating the physical properties of Arctic snow (vertical density profiles, stratigraphy, etc.) as demonstrated by many past studies (Barrere et al (2017), Gouttevin et al 2018 and Royer et al 2021). In our case, this is shown by comparing an observed density profile with the density profile simulated by Crocus (see Supplementary Figure 1).

[Figure]

Supplementary Figure 1: Profiles of snow density (blue) and thermal conductivity (orange) at the 22 March 2019. The simulated density profile at the same time is also shown.

**13.** Line 156: I would not classify a temperature error as a percentage. Is that in °C or K? I suspect that a percentage error for thermocouples would refer to the temperature difference between the thermo-junction in the snow and the reference temperature junction.  An error in the accuracy of the reference temperature thermistor would likely be expressed in fractions of a degree Celsius over a range of temperatures.

We agree. We changed our description of the error (modifications in bold):

"[...]while the temperature measurements **of the type-T thermocouples have an accuracy of 0.5°C in the temperature range under study**."

**14.** Line 176: Again, a percentage error in a temperature is difficult to interpret.

Please see comment above.

**15.** Line 178: Is the error for precipitation 0.15 mm per half-hour, 0.15 mm per precipitation event, or 0.15 mm per time interval during which precipitation was recorded? Okay I looked it up. Accuracy is specified as 0.1% of full scale, and 0.15 mm is the repeatability, while sensitivity is 0.1 mm. If the 1500 mm version of this gauge was employed, then the accuracy is 1.5 mm over a season, based on what actually entered the gauge and closer to 0.15 mm per event, but this does not account for snow undercatch caused by wind deflection around the gauge. Were the snowfall data corrected for undercatch of snow. There are equations available for correcting this gauge with a single Alter shield based on wind speed at the gauge height. Smith (2006) found that a Geonor T200B with a single Alter shield caught only 36% of the snow caught by a Double Fence Intercomparison Reference (DFIR) gauge (the WMO standard) in Bratt's Lake Saskatchewan. Were the precipitation gauge data corrected for undercatch, and if so, which equation was employed?

Yes, the precipitation data were corrected for undercatch. We used the equation from Kochendorfer et al. 2017. The following sentence was added to the manuscript:

"**Precipitation data were corrected for undercatch of solid hydrometeors using the transfer function of Kochendorfer et al. (2017),  which depends on wind speed and air temperature.**"

Concerning the error for precipitation, in Kochendorfer et al. (2018), they report an RMSE of 0.25 mm for the 1500 mm gauge used here,  after correcting the data for undercatch. Thus, we feel that this is the best way to state the error, as it includes both the accuracy of the gauge and the data post-processing. Note that we initially reported an error of 0.15 mm, which was incorrect and was corrected in the next version.

**16.** Figure 2: Do the tick marks for each month represent the start of the month or the mid-point? It appears to be the start.

To clarify this, the following sentence was added in the figure caption: "Labeled ticks marks on the x-axis indicate the start of each month."

**17.** Regarding discussion of sensible and latent heat fluxes: Rather than using the terms 'increases' and 'decreases', it may help to include direction, such as strong upward or strong downward or weak upward or weak downward fluxes.

Good point. Indications of the direction of the fluxes were be added in the discussion on the sensible and latent heat fluxes.

18. Line 298-302: Errors in the simulated snowpack albedo could cause differences in daytime net radiation, and this could be checked and plotted, although I suspect there is more error in the longwave component. A phase shift may be a result of poor thermal conductivity simulations or issues related to simulated fluxes and stability corrections. Figure 8 would be easier to interpret and would be more informative if air temperature and the radiative skin temperature of the snow surface (based on outgoing longwave radiation) were also plotted. QG while not at the surface, could inform Figure 8.

The modeled albedo was in the same range as the observed albedo with only slight differences (mean difference of 0.0006 for winter 2017-18). Such small differences do not strongly affect the energy balance in winter as the incoming shortwave radiation is not that high. Thus, we agree that the longwave component is more important here.

[Figure]

Figure: Observed and modeled albedo during winter 2017-18.

We added a note to the manuscript describing the difference of the modeled and observed albedo:

"**During the period shown in Figure 8 and throughout the study period, the modeled albedo was always in the same range as observations (differences <0.05). The mean modeled albedo was 0.01 greater than observations.**"

  Furthermore, since thermal conductivity depends on snow density, the fact that the density profile is not modeled correctly certainly brings its own set of errors, as shown in several other studies  (e.g., Barrère et al. 2017; Royer et al. 2021).Thus, part of the phase shift could be due to this error in thermal conductivity. A note stating this possibility was added to the manuscript:

"At certain moments, a few hours of phase-shifting between the residual and the snow heat flux can be observed, **which might be due to an inaccurate simulation of the snow thermal conductivity.**"

As requested by the reviewer, we added the air and surface temperatures to Figure 8.

[Figure]

Figure 8: Comparison between observed and modeled hourly means of all constituents of the energy budget at the snow surface from March 14 to 23 2020. The residual snow heat flux is obtained by subtracting the turbulent heat fluxes from the net radiation. **Also shown are the air and surface temperatures during this period.**

19. Figure 9: This figure would be easier to interpret if air, snowpack and soil temperatures (and snow surface radiative temperature) were plotted as points or lines. Early in the season, the sensible and soil heat fluxes to the snowpack are not enough to balance the radiative losses. Soil temperatures would provide information about when the soil water is freezing, which represents an energy source under the net radiometer although below the snow/soil interface. The differences between soil and snowpack temperature and the timing of soil freeze are important pieces of information that would help to interpret what is happening. Given that the soil heat flux is supposed to represent the flux at the ground surface, or in this case at the soil/snow interface, what sort of values for QG would be obtained by using the gradient between the lowest snow temperature and the 4 cm soil temperature with an average thermal conductivity?

We added the requested temperature in a third panel under Figure 9. However, we feel it is now a bit overloaded and thus, we put it in the supplementary material and left the original version in the manuscript.

[Figure]

Figure 9: a) Observed and b) simulated daily snowpack energy budget terms comprising sensible (QH) and latent heat fluxes (Qe), net radiation (Q*), ground heat flux (QG, and the change in the internal energy of the snowpack dU/dt, during the first half of winter 2018/19. The modeled ground heat flux also includes the heat storage change dU/dt in the snowpack. The modeled snow enthalpy change is not shown because the modeled enthalpy includes changes due to precipitation and is therefore not comparable to observations. In the lower panel c) the temperatures of the air, the surface, the snow and the ground are shown.

We do not recommend using one temperature measurement in the snow and one in the ground as the thermal conductivity between these two varies heavily. A test using an average thermal conductivity between the two media revealed inconsistent results.

20. Line 409-414: Flow separation and drainage flows may be a factor at this site. Could a temperature profile be examined using the radiative temperature of the snowpack surface, the air temperature from the tower, and any unburied snow temperature sensors? It might give some idea of the level of stratification at night. Radiation errors might prevent the use of unburied snow temperature sensors during the day.

The atmospheric stratification is already quantified via the bulk Richardson number calculated between the snow surface and the measurement level of the EC sensor, a few meters above the snowpack. Using this approach allows us to have reasonable measurements of the $Ri_b$ at all times, which would not be the case if we were to use unburied thermocouples that get heated by the sun during the day (as you are pointing out). A more detailed analysis of turbulent heat fluxes in winter and their parameterization for land surface models, in particular in relation to atmospheric stratification, is under way. For this current paper, we refer to Table 1 showing the stability regimes.

21. Line 415-418: It would be interesting to see the heat fluxes calculated at the bottom of the snowpack, compared with those calculated in the soil column (and those using the 4 cm soil temperature and the lowest snow temperature).

We added a figure showing the heat fluxes in the snow at 7 cm and the ground heat flux in the supplementary material. The heat flux using one temperature measurement in the snow and one in the ground are not shown (see comment 15).

[Figure]

Supplementary Figure 5: Soil heat fluxes computed at a depth of 7 cm (orange) and snow heat flux computed 7 cm above the soil surface. Note that the large negative soil heat flux on 4 April corresponds to a rain-on-snow event. Positive values indicate fluxes from the soil to the atmosphere.

**Corrections and minor suggestions:**
We made all the minor changes requested in comments 22 through 26, and 28 through 31.

22. Line 14: I would change "freezing air temperatures" to "air temperatures less than 0°C".
23. Line 15: I am not sure that "constraints" is the right word here. Perhaps "challenges".

24. Line 30: Just to be precise, I would state "…and QG is the ground heat flux at the snow/soil interface".

25. Line 38: Change "snow" to "snowpack".

26. Line 53: Since the term "sublimation" can refer to the conversion of water vapour to ice, even though the authors use "condensation" for this process, I would change the wording from "However, according to Liston and Sturm (2004), sublimation in the Arctic can make up as much as 50% of the total winter precipitation" to "However, according to Liston and Sturm (2004), sublimation losses in the Arctic can deplete as much as 50% of the total winter precipitation."

27. Line 186-7: Do the authors mean that when there is snow on the ground in the model, the surface is always 100% covered with snow, as opposed to a fractional cover based on SWE or depth?

Yes, the ground is assumed to be completely covered with snow when the snow height exceeds 1 cm. Otherwise, the surface is considered to be covered to a certain fraction by snow, while the rest is snow-free. Consequently, the albedo and the turbulent fluxes are calculated separately for the snow-free and the snow-covered part. This, however, leads to very different values of the albedo and the turbulent fluxes compared to observations. For this reason, we have chosen not to use a fractional snow cover.

We clarified this in the manuscript:

"We also used the option in Crocus that allows for the surface to be 100% covered with snow **once the snow height exceeds 1 cm**."

28. Line 225: This sentence is written with the assumption that the reader is familiar with the low pressure systems in the region. I would reword it as: "Snow usually accumulates quickly in the fall as precipitation events are more frequent due to the large low pressure systems which are prevalent at that time of year."

29. Line 226: Change "rates drops" to either "rates drop" or "rate drops".

30. Line 362: Change "recomnd" to "recommend".

31. Line 369: Change "sublimation accounts for only 5% of winter precipitation" to "sublimation losses represent only 5% of winter snowfall".

**Responses to Reviewer #2**

The manuscript "On the energy budget of a low-Arctic snowpack" by Lackner et al. presents measurements and modeling of the surface energy balance for a site near Umiujaq, Canada. The authors evaluate a unique data set on the snow surface energy balance including measurements of turbulent fluxes with the eddy covariance technique. The manuscript is well written and I recommend publication in TC following revisions.

**Major comments:**

1. I am missing a dedicated section on the observed snow pack structure, including density and grain size profiles, as well as presence/absence of depth hoar, wind slab and melt layers. I am aware that such snow profiles exist only for a few points in time, but they are nevertheless important for the understanding of the snow pack processes. I am also missing an evaluation of CROCUS simulations against these snow profiles which in my view is critical for understanding model performance. In CROCUS, the snow thermal conductivity is directly related to snow density, which again is controlled by snow microstructure, wind compaction, melt, etc. Therefore, the heat flux into the snow pack and the heat storage within the snow pack are strongly related to the simulated density profile.

Thank you for recommending our manuscript for publication. We highly appreciate your helpful comments that allowed us to significantly improve the paper.

Yes indeed, information about the internal structure of the snow is important. For this reason, a second paper about this is in preparation and will be submitted soon. There, we compare the snowpack properties such as density, thermal conductivity, grain types, and snow temperature at this very site to simulations of Crocus. In order to avoid repetition, we did not include a section on this topic in this paper. However, following your concerns that some information is needed to understand the snowpack processes, we included some figures in the supplementary material. As a density profile representative for this site is already present in the supplementary material, we added a profile of the grain types and snow thermal conductivity as well as a simulated snow density profile (see figures below).

Concerning the evaluation of Crocus against these profiles, as mentioned above, this is the subject of another complementary paper, which examines this issue in much more detail. It has already been shown on a few instances that Crocus is not able to reproduce Arctic snow density profiles, and this study is no exception. We added a note in section 2.4.2 that the internal properties of the snowpack were not well simulated by the model as observed in other studies:

"Similar to previous studies where Crocus was run in an Arctic setting (Barrere et al. (2017) and Royer et al. (2021)), the density profile simulated by the model did not match the observations. While the observed vertical snow density profile was rather constant with height, Crocus showed a strong decreasing trend towards the snow surface."

[Figure]

Supplementary Figure 1: Profiles of observed snow density and thermal conductivity on March 22, 2019 and the corresponding simulated density profile. The stratigraphy on the same day is shown on the right.

[Figure]

Supplementary Figure 2: Stratigraphy on March 22, 2019 corresponding to the density and thermal conductivity profiles shown in Figure 1.

Along the same lines, there should be results and a discussion on the role of the ground below on the snow energy balance. The authors refer the reader to Lackner et al., 2021, for a more thorough description of the ground properties, but there are many critical aspects that the reader needs to know, for example: Is there permafrost at the specific location of the measurements? If yes, what is the active layer thickness, if not, what is the thickness of the seasonal frost layer? Is there a water table on top of the permafrost, which first has to refreeze in fall/early winter, thus confining ground surface/snow base temperatures to close to zero degrees during this time? What is the difference between the ground surface temperature and the snow surface temperature which defines the overall temperature gradient over the snow pack? It should be possible to check all these aspects in both the measurements and the model. Some of the points raised might help explain the discrepancy in Qg (L. 332).

There is scattered permafrost in the valley in the form of lithalsas, but our very site is free of permafrost. We clarified this in the manuscript:

"Permafrost is discontinuous to sparse and is rapidly degrading (Fortier et al. 2011). **At the precise location of the experimental setup, no permafrost was present.**"

The Figures shown below depicting the temperatures of the ground and the snow surface are going to be included into the supplementary material for winters 2018-19 and 2019-20. The deepest soil temperature measurement is at 50 cm depth (see figure below). The temperature roughly varies between –10°C in winter and +10°C in summer. Thus, the freezing depth is probably a few meters deep.

[Figure]

Supplementary Figure 3: Temperatures of the ground (at 14 cm depth), the snow (at 17 and 45 cm height), and at the snow surface during winter 2018-19.

[Figure]

Supplementary Figure 4: Temperatures of the ground (at 14 cm depth), the snow (at 17 and 45 cm height), and at the snow surface during winter 2019-20.

[Figure]

Figure: Soil temperature at 50 cm depth.

**Specific comments:**

1. Fig. 1: A site map would be nice, which for example shows the distance to the coastline.

An inset map was added to Figure 1. We specified that the site is 4 km from the Hudson Bay coast and 4 km from lake Tasiujaq.

[Figure]

Figure 1: Upper panel: Study site with a) the main 10-m flux tower with the eddy covariance setup, b) a precipitation gauge, c) a 2.3-m high mast hosting the 4-component radiometer and d) a vertical pole holding an array of thermocouples and heated needles. The inset map shows the location of the site in the Tasiapik valley, some 4 km east of Hudson Bay. Lower panel: Schematic of the study site illustrating the main instruments monitoring energy balance terms. The whole experimental setup is contained within 20 m.

2. L30: Consider adding a clarification to Qg, like "…, i.e. the energy flux through the snow-ground interface".

We added the recommended clarification:

" $Q_G$ is the ground heat flux, i.e the energy flux through the snow-ground interface."

3. L75: I recommend "Here, we measure…" instead of "Here, we attempt…". You've done it after all, despite the obvious difficulties!

Yes, indeed "Here, we measure" is more appropriate. We changed it, thank you.

4. L129: How often was the gap filling needed, i.e. what overall fraction of the data set is not the original measurements?

The overall percentage of gap-filled data is 44% for the sensible heat flux and 61% for the latent heat flux. Note that this percentage includes also the longer periods of instrumental failure in winter 2017/18 and 2018/19 visible in Figure 4. We added the following statement to clarify this:

"Including longer power outages, gaps were present for 44% of the study period for sensible heat fluxes and 61% for latent heat fluxes."

5. L139: 1W/mK seems very low, this would correspond to a rather dry soil. If the soil pores were largely ice-filled, a thermal conductivity of >2W/mK would be more appropriate. How does this assumption affect the computed heat fluxes and the comparison to the model?

Indeed, the value for thermal conductivity of the soil used here is rather low.  The span of possible values is rather high in the literature and thus, we compared the heat flux in the ground to the one measured in the snow just above the soil, as detailed in the manuscript. With a value of 1W/mK, the two fluxes match in magnitude. A value of 2 W/mK or higher would at least double the heat flux in the ground and it would therefore be far higher than the measured heat flux in the bottom of the snow and the modeled soil heat flux. For this reason, 1 W/mK seemed the most appropriate choice. There is a reasoning supporting the selection of  this value. The water content of this sandy soil is very low, as the large grains and the large pores retain little water. Therefore, there is probably little ice, leading to a lower thermal conductivity than expected for frozen soils with smaller grains.

We added a corresponding note in section 2.3.2:

"Note that we compared the resulting ground heat flux to measurements of the snow heat flux 7 cm above the soil to validate this value of thermal conductivity."

6. L156: What does an error of 0.75% mean for temperature? In which unit is temperature referred to here?

We changed the specification of the error to °C :

"[...]while the temperature measurements **of the type-T thermocouples have an accuracy of 0.5°C in the considered temperature range**."

7. L158, Sect. 2.4.1: Briefly describe the physics of the ground module that is used in the simulations and as such provides the lower boundary for the CROCUS model. Some of the text from l. 179 could be moved to this description.

We added a brief description of ISBA, the soil model coupled to CROCUS:

"The soil and vegetation model ISBA is coupled to Crocus and simulates all water and energy exchanges between the different soil layers and with the snowpack above the ground. For this purpose, the one-dimensional Fourier law and a mixed-form of the Richards equation are solved explicitly (Boone et al. 2000; Decharme et al. 2011). The characteristics of the vegetation are selected from a list containing 19 different vegetation types (ECOclimap; https://opensource.umr-cnrm.fr/projects/ecoclimap-sg/wiki) using the site coordinates, or alternatively they can be specified by the user. In this study, the latter option was used."

8. L284: Please introduce Q* again for clarity. It is not used in the paragraph on net radiation, so readers have to go back to the initial sections if they are not familiar with the symbol.

Good point. We changed the sentence to:

"Daily means of the **turbulent heat fluxes** $Q_H$, $Q_E$, and **the net radiation** $Q_*$ time series simulated by ISBA-Crocus [...]."

9. L315: Briefly state what the Qa with the arrow means.

A clarification after the equation was added:

"[...], **where the advective heat input Qa is neglected.**"

10. L332: Here, an uncertainty analysis on the different factors used to calculate Qg, especially the thermal conductivities, could help. See also major comment 2.

Given the fact that the thermal conductivity was already chosen on the lower bound of the possible values, it is very unlikely that the discrepancy between the observations and modeled values can be attributed to uncertainties of the observations, as observations are already much higher than modeled values. We added a sentence mentioning the comparison between the heat flux in the bottom of the snow pack and the ground (see comment 5).

11. L339: Switch order of references.

 The order was switched.

12. L364: Consider adding a statement on the timescales. The authors write themselves that the model is doing better for longer periods so that some applications of the model may be less compromised than this statement suggests.

A statement on the time scales was added to the sentence:

"Meanwhile, further studies are underway at the site targeted in this study to evaluate optimal turbulent heat flux parametrizations, **particularly for sub-daily time scales.**"

13. L365: I am missing three aspects in the section on sublimation and drifting snow. First, the percentage of SWE lost from sublimation also depends on snowfall/total SWE, so this aspect should be considered when comparing to previous studies (L. 370). Second, the authors should at least qualitatively comment on the intensity of the snow drift events. The daily average wind speeds (Fig. 2) seem fairly low and only marginally above the limit for snow drift of 5-6 m/sec, as e.g. assumed in

Crocus. In particular, prolonged storms with wind speeds >>10m/sec where snow drift is much more intense seem to lack completely. If correct, this could partly explain why the measurements do not show a higher sublimation. Finally, blowing snow events do not strongly change the constraints on energy availability and humidity that also apply for sublimation from flat surfaces, as mentioned in the manuscript. For very cold air and snow temperatures, for example, the humidity at saturation and thus the potential vapor deficit of the air are small which limits the latent heat fluxes and thus sublimation. The same is true for very moist air.

Thank you for the comment. It is true that the percentage of mass lost due to sublimation depends on total snowfall which is rather high at our site. We added a corresponding phrase to the manuscript:

**"On the other hand, the winter precipitation is quite high at our site, which naturally decreases the fraction of sublimation losses to precipitation."**

In L. 372 we state that blowing snow events are observed several times per week on time lapse cameras as the snow height decreases during these periods. Unfortunately, we cannot further specify the amount of snow blown away by the wind as we did not have instruments for this.

Right, blowing snow does not strongly change the constraints from the water vapor deficit and the air temperature on sublimation but as detailed in the manuscript, Mann et al. (2000) showed that due to the high density of snow particles in the air, it can become almost fully saturated. This represents conditions different to those found above a flat surface. Sublimation during blowing snow is still an active area of research with large uncertainties.

14. L416: This is an important finding which inspires confidence in the results and should thus be presented in more detail in the Results section. See also my comment L. 139.

A sentence highlighting the comparison between the heat fluxes in the bottom of the snow and the ground was added to the manuscript in the results section. See comment 10.

15. References: the doi link to Lackner et al., 2021, points to a different paper

Well noticed! The correct doi link is the following (doi.org/10.1175/JHM-D-20-0243.1) and it was integrated in the new version of the article.

**Responses to the reviewer #3**

**Overall comments:**

Overall this paper should be published. There are earlier papers looking at snow energy balances in the subarctic/low arctic (I refer to Anthony Price' early work in the Schefferville area when he was a PhD student at McGill). In the Lackner et al paper there are notable discrepancies between the real world and the model. Though I thought the discussion was quite good it might be an idea to add a few words directed at the model and why it doesn't seem to model Qh well. There is acknowledgment of this but I wondered if the authors, after their experience with this data set have suggestions to better the model? I have made a few comments below some out of curiousity and others more specific. In tundra environments herbs and shrubs in the snowpack can play a role in the energy balance at I assume they can...especially in the late winter when the sun is becoming more intense and in the early spring as they absorb radiation (I understand coniferous plants can photosynthesize under the snow surface).
Overall I would accept the paper with some minor modifications.

Thank you for your positive comments on our manuscript.

Crocus depends on complex relationships between snow, atmosphere, vegetation and soil. Since the model has been little compared to observations of the surface energy balance in the Arctic environment, we believe that the first logical step, before any proposal for improvement, is to perform a rigorous evaluation of its performance highlighting its strengths and weaknesses. This is thus the main objective of the modeling part of the paper.

It should be noted that some of our instruments (e.g. snow temperature measurements) were deployed in a shrub-free zone, thus preventing us from targeting their impact on snow properties. Also, all the shrubs in the valley are deciduous, so there is no photosynthesis in winter. Finally, we wish to remind the reviewer that we excluded spring from our analyses, where the contribution of shrubs to e.g. albedo is certainly more important than in wintertime.

Below, we answered all your comments and have proposed modifications when requested.

**Specific comments:**

1. Page 3: line 50: they measured ground heat flux under the glacier? Is that right?

No, at the glacier site the ground heat flux was not measured.

2. Line 53: small point: would be slightly clearer to say that 50% of the winter precip is lost to sublimation

Indeed, thanks. We changed the wording to:

"[...], **water losses due to sublimation** in the Arctic can make up to 50% of the total winter precipitation."

3. Line 59: though in the subarctic, the earlier study by Price (PhD at McGill) was a detailed energy budget of a snowpack. (Water Res. Res Vol 12:4: 686-694)

Thanks for the reading suggestion, it is indeed a very interesting article. Unfortunately, we believe it is best not to cite it here for a few reasons:
- the authors are interested in a forest site and here we are aiming for a tundra;
- the period under study is spring whereas here we are targeting winter;
- not all components of the energy balance are measured and therefore there are no conclusions on the turbulent heat fluxes, which occupy a prominent place in our article.

4. Page 7: line 145 or so: as the density of the snowpack changes the amt of air space changes...would this not have an impact on calculating the heat capacity of the snowpack?

Good point, thanks. Only the heat capacity of the snow (ice) was used for the calculation here, as the air space and the associated heat capacity only make up a negligible fraction of the one of ice.

5. Page 8: Line 179: 12 m or 1.2m? earlier you state that you have thermocouples at 4 and 14 cm...so not sure what you are doing here...assume this is a typo

Yes, the thermocouples were installed at 4 and 14 cm depth, but the 12 m soil column refers to the model. Here, a thick soil column is needed in order to also include heat storage effects of deeper soil layers. No measurements are present at these depths but as stated in L. 183, a spin-up of 5 years was used to obtain an equilibrium of the thermal regime in these deep layers.

6. Line 180: did you measure field capacity in the lab

No, unfortunately no such measurements were performed. However, we used time series of the soil water content at several depths to estimate the field capacity and the saturation water content. The field capacity was then taken as the values of the soil water content some time after rain events.

7. Line 194: a small question (though it probably makes little difference) do you adjust specific heat wrt temp – I assume you are but would it make much difference?
Thinking here too about air in the snowpack

 L. 194 refers to the calculation of the turbulent fluxes in the model Crocus and to the best of our knowledge, the specific heat of the air is not adjusted for temperature variations. But as you mentioned, it should not make much difference. In the snowpack the air space is not included as detailed in comment 4.

8. Figure 2: Probably quite explainable...however there are places here where your snowpack drops significantly over what appear short periods of time...wind? (coupled with compression) -for example 2018-19 late Jan there is a snowfall (i assume) and on or about 7 FEb there is a sudden drop of snowpack height from about 55cm to 40cm (or so)/ as well in 2019-20 late Feb snowpack is about 75cm or so and by mid March

around 60cm...significant wind at this time..assume it is wind scour

Yes, in fact blowing snow is a common phenomenon at the site and wind transports the snow to a site further down the valley where snow height is significantly higher. In L. 371/372 the approximate frequency of blowing snow is stated:

"[...] blowing snow events, which frequently occur at the site (observed several times per week on time-lapse images) [...]"

 A paper comparing the snow heights of this site with another one further down the valley is in preparation. There we plan to study this phenomenon closely.

9. p10. Line 255: I assume that this pattern of precip is tied somewhat into the proximity of Hudson's Bay....does the drop in precip in December tie into ice covering
a large part of the bay?

 We also think that this precipitation and wind pattern are strongly influenced by Hudson Bay. As stated in L.222, Hudson Bay freezes around mid-December and subsequently wind speed and precipitation rates drop. Thus, one can assume that this is due to the freezing of Hudson Bay. We added this to L. 255:

"Just as for the wind speed, precipitation rates **are presumably also influenced by Hudson Bay and** drop at the end of December and remain rather low until March, [...]."

10. Figure 3: net radiation in 2018-19 in early January show a slightly positive balance out of curiousity what is happening here; similarly 2019-20 in early March; in both cases longwave in and out is balanced ...significant cloud cover? Thinking that in years ahead with more of Hudson bay staying open longer there may be increased cloud cover...might be interesting to speculate how this may play a role in the energy budget of these low Arctic snowpacks?

A significant cloud cover in combination with relatively warm air temperatures (still below 0°C) was responsible for the peaks in net radiation. There might be an effect of climate warming and Hudson Bay being open for longer periods on cloudiness but our site is already very cloudy so it is hard to speculate in this topic based on the experiences from this site.

11. Figure 6: you are inferring a linear relationship here...is there any point? are these relationships significant? What might be interesting is to look at (for example) in (b) at Ts-Ta 1°C to about 2.3°C the range of Qh is very large, though focused primarily between ~+50 and -45 W/m2...for more or less similar Ts-Ta values you get a very large range of Qh: is there anything of interest here: similarly for (d) between ~.15VPD and .18 VPD a very large Qe range

In many equations used to calculate the turbulent fluxes in land surface models, wind speed, Ts-Ta and VPD enter linearly, as can be seen in equation 6 and 7 in the manuscript. Thus, here we wanted to compare those quantities with observed fluxes and look whether the linear dependence can be observed.

12. P 14 line 280: are these relationships statistically significant? Though when looking at these relationships the important thing is the visual message that the model in Qh and Qe

under/overestimates for a reasonable range of the W/m2 range

We agree, the importance of Figure 7 is the visual message that there are some issues with the performance of the model over the observed heat flux range. We therefore did not test the statistical significance of the model.

13. P15 Line 301-302
You refer here to residual snowpack heat flux. In our experience in subarctic and low arctic snowpacks there can be a notable concentration of coniferous shrubs that absorb energy and appear to be photosynthesizing (we were not measuring this but colleagues mention this goes on). Is the snowpack in any way impacted by energy absorption by shrubs in the snowpack at all? Seems it might...of course depending on the characterisitics and density of the shrubs. I see no mention here of shrubby veg...so assume this isn't the case here?

The occurring shrub types here, mainly dwarf birch (compare L. 84-86), are all deciduous shrubs, so there is no photosynthesis in winter. The shrubs probably have an impact by absorbing radiation, which then heats up the snowpack. This, however, is more crucial in spring (April, May and June) when the shortwave radiation is much larger and the twigs of the shrubs are closer to the snow surface or even stick out of the snowpack. Furthermore, a significant bending of the shrubs was observed in the snow pits, which were dug in areas covered by shrubs.

14. P 15 line 294: would the sensors be in any way impacting the energy balance? as they are close to the
surface of the snowpack/ what about blowing snow along the surface...impact the ability of the sensors in any way?

Indeed, it is hard to measure the temperature of the snow close to the surface as the sensors may suffer from solar loading. We discuss these limitations in lines 393 to 397:

For this reason, we used only measurements where the sensor had a certain distance (3-5 cm) to the snow surface. Also, the sensors were enclosed in a white tubing to minimize radiative effects, as shown in Figure 1. But obviously certain influences cannot be ruled out making the snow heat flux very hard to measure accurately.

15. Figure 7. so with Qh you have wide scatter in the model. might be interesting to look at some observed values of Qh and investigate the very large range of modelled responses...to isolate what other variables might be playing a role in producing a large range/
for example at ~+10W/m2 (observed) you have a range of modelled responses between ~ -5 and +35
....in the upper range of the modelled responses versus the lower range were there any standout differences in other variables (wind speed?)

Atmospheric stratification is a very complex subject that is not very well captured by the model and consequently it is the largest source for the scatter in Figure 7. Also, we checked some variables such as temperature and wind speed and they already showed substantial scatter for observed values around 10 W m-2 of Qh, so no variable could be identified that causes the large scatter in the model.
The same is true for very high modeled values.

16. Figure 7 and Figure 8

I see why you have included Fig 8 but there are some interesting differences between these figures which leads to a couple of questions. 1. In Fig 8 where you are confident you have good control on all measureable variables there is very good agreement between the modelled and observed Qe. However in Fig 7 there is consistent underestimation (I think I am reading this correctly). So what did you learn (and what could we learn) from this discrepancy with respect to the model and where the issues are?

2.With respect to Fig 7 if it wasn't for a number of values >25 W/m2 or below -5 W/m2 the relationship could well be close to vertical as I would guess 90+% of the values fall within the large lump. In Fig 8 the model again overestimates under very good controlled conditions. Any idea why the model seems to overestimate sensible heat? Any physical explanation?

1. During the period shown in Figure 8, the conditions were favourable for the measurements, including no issues related to blowing snow. Thus, our confidence in the measurements is higher. But most importantly, in Figure 7 the bulk of the point is close to 0 (between $-10$ W m$^{-2}$ and 5 W m$^{-2}$) thus, even with the underestimation of the model, the absolute difference between the observations and simulations is in the order of a few W m$^{-2}$, so rather small. This small difference is visible in Figure 8. We have chosen to present all fluxes in Figure 8 within the same range, unfortunately in the case of Qe this choice negatively affects the ability to compare modeled and observed Qe in more detail, but the differences are in the same range as the ones presented in Figure 7.

2. Even for the bulk of the points the relation between observed and modeled is not vertical, in fact, for the range between 0 and 10 W m$^{-2}$, the correlation is quite good with a slope close to 1 as indicated by the black line in Figure 7. For higher values, there is a substantial overestimation, it seems that the snow surface temperature difference between the model and the observations tends to be higher than the average difference. Thus, a possible reason might be that the simulated snow surface temperature is a cause for the high values.

17. P17. Line: 315: I assume crossing out the Qa term simply means that while you identify it is important it is not included here...the arrow to the zero?

The arrow over Qa means that although we acknowledge the advection to be part of the energy budget, we assume it to be sufficiently small to be neglected. And even in periods where this term is higher, we cannot measure it and have to neglect it. We added a statement clarifying the meaning of the arrow:

"[...], **where the advective heat input Qa is neglected.**"

18. P17 Line 326: so you have more energy coming in than can be accounted for in heat loss from the system/ possibilities? (a bit of guess work here)underestimating energy used to raise the temperature of the ice in the snowpack to melting point? Early in the melt as melting water infiltrates the frozen ground (is it?) the runoff refreezes complicating the issue? Some issue with calculating heat loss from the snowpack re: turbulent fluxes.....and going out on a limb here but are there periods when you may have laminar flow and underestimating heat loss// later I see that isn't an issue...but thought I would leave this comment in

It is very hard to say where we miss energy fluxes. Issues related to melt can be ruled out as we did not include periods when melting occurred. In fact, this was an important reason for the choice to leave out these periods. Turbulent fluxes are a probable source for missed energy. Even in summer, most energy budget studies report unclosed energy budgets where more energy is coming in than going out. During

the winter conditions here, measuring the turbulent fluxes is even more challenging with the atmospheric stratification being mostly stable (see Table 1).

19. P17 line330: the phrase: "slightly less"... are you being generous using this term especially for Qh? re: fig 7

We changed the wording here to:

"[...] is well simulated, but this is less true for [...]"

20. P17 L333: issue re: observations of instrument location an issue...

The location of the instrument does change the measured ground heat flux, but given the large discrepancy between the observed and modeled ground heat flux the location of the instrument is likely not the dominant factor here.

21. P 19: line spelling of recommend

Thanks, we corrected this typo.

22. P20 Line 397: again, if herbs and shrubs are present are they playing a role here?

For the measurement of the snow heat fluxes at this site the shrubs don't play a role as the instrument was placed on pure lichen (compare L. 112).

23. P21 line 436: spelling of word: consisted

Again, thanks for the remark. We fixed the typo.

**References**

Barrere, M., Domine, F., Decharme, B., Morin, S., Vionnet, V., and Lafaysse, M.: Evaluating the performance of coupled snow–soil models in SURFEXv8 to simulate the permafrost thermal regime at a high Arctic site, Geosci. Model Dev., 10, 3461–3479, https://doi.org/10.5194/gmd-10-3461-2017, 2017.

Gouttevin, I., Langer, M., Löwe, H., Boike, J., Proksch, M., and Schneebeli, M.: Observation and modelling of snow at a polygonal tundra permafrost site: spatial variability and thermal implications, The Cryosphere, 12, 3693–3717, https://doi.org/10.5194/tc-12-3693-2018, 2018.

Boone, A., Masson, V., Meyers, T., & Noilhan, J. The influence of the inclusion of soil freezing on simulations by a Soil–Vegetation–Atmosphere Transfer Scheme, J. Appl. Meteorol., 39, 1544-1569, doi: https://doi.org/10.1175/1520-0450(2000)039<1544:TIOTIO>2.0.CO;2, 2000.

**Decharme, B., Boone, A., Delire, C., & Noilhan, J. Local evaluation of the interaction between Soil Biosphere Atmosphere soil multilayer diffusion scheme using four pedotransfer functions,** *J. Geophys. Res.*, 116, D20126, doi:10.1029/2011JD016002, 2011.

**Fortier, R., LeBlanc, A.-M., & Yu, W.. Impacts of permafrost degradation on a road embankment at Umiujaq in Nunavik (Quebec), Canada.** *Can. Geotech. J.* 48(5): 720-740. https://doi.org/10.1139/t10-101, 2011.

**Kochendorfer, J., Nitu, R., Wolff, M., Mekis, E., Rasmussen, R., Baker, B., Earle, M. E., Reverdin, A., Wong, K., Smith, C. D., Yang, D., Roulet, Y.-A., Meyers, T., Buisan, S., Isaksen, K., Brækkan, R., Landolt, S., and Jachcik, A.: Testing and development of transfer functions for weighing precipitation gauges in WMO-SPICE**, Hydrol. Earth Syst. Sci., 22, 1437–1452, https://doi.org/10.5194/hess-22-1437-2018, 2018.

**Lackner, G., Nadeau, D. F., Domine, F., Parent, A., Leonardini, G., Boone, A., Anctil, F., & Fortin, V. (2021). The Effect of Soil on the Summertime Surface Energy Budget of a Humid Subarctic Tundra in Northern Quebec, Canada**, *Journal of Hydrometeorology, 22*(10), 2547-2564. doi: https://doi.org/10.1175/JHM-D-20-0243.1

**Mann, G. W., Anderson, P. S., and Mobbs, S. D. Profile measurements of blowing snow at Halley, Antarctica,** *J. Geophys. Res.*, 105 (D19), 24491– 24508, doi:10.1029/2000JD900247, 2000.

**Royer A, Picard G, Vargel C, Langlois A, Gouttevin I and Dumont M (2021) Improved Simulation of Arctic Circumpolar Land Area Snow Properties and Soil Temperatures**. *Front. Earth Sci.* 9:685140. doi: 10.3389/feart.2021.685140

s